# Mitochondrial arginase-2 is essential for IL-10 metabolic reprogramming of inflammatory macrophages

Jennifer K. Dowling [1,2,3,4,10], Remsha Afzal [1,10], Linden J. Gearing [2,4], Mariana P. Cervantes-Silva[1], Stephanie Annett[1], Gavin M. Davis[5], Chiara De Santi[1], Nadine Assmann[5,6], Katja Dettmer [7], Daniel J. Gough [4,8], Glenn R. Bantug [6,9], Fidinny I. Hamid[4,8], Frances K. Nally[1], Conor P. Duffy[1], Aoife L. Gorman[5], Alex M. Liddicoat[5], Ed C. Lavelle [5], Christoph Hess[6,9], Peter J. Oefner[7], David K. Finlay [5], Gavin P. Davey[5], Tracy Robson [1], Annie M. Curtis [1], Paul J. Hertzog[2,4], Bryan R. G. Williams [4,8] & Claire E. McCoy [1,3,4,8✉]

Mitochondria are important regulators of macrophage polarisation. Here, we show that arginase-2 (Arg2) is a microRNA-155 (miR-155) and interleukin-10 (IL-10) regulated protein localized at the mitochondria in inflammatory macrophages, and is critical for IL-10-induced modulation of mitochondrial dynamics and oxidative respiration. Mechanistically, the catalytic activity and presence of Arg2 at the mitochondria is crucial for oxidative phosphorylation. We further show that Arg2 mediates this process by increasing the activity of complex II (succinate dehydrogenase). Moreover, Arg2 is essential for IL-10-mediated downregulation of the inflammatory mediators succinate, hypoxia inducible factor 1α (HIF-1α) and IL-1β in vitro. Accordingly, HIF-1α and IL-1β are highly expressed in an LPS-induced in vivo model of acute inflammation using $Arg2^{-/-}$ mice. These findings shed light on a new arm of IL-10-mediated metabolic regulation, working to resolve the inflammatory status of the cell.

[1] School of Pharmacy and Biomolecular Sciences, Royal College of Surgeons in Ireland, Dublin 2, Ireland. [2] Centre for Innate Immunity and Infectious Diseases, Hudson Institute of Medical Research, Clayton, VIC, Australia. [3] FutureNeuro, SFI Research Centre, Dublin 2, Ireland. [4] Department of Molecular and Translational Science, Monash University, Clayton, VIC, Australia. [5] School of Biochemistry and Immunology, Trinity College Dublin, Dublin, Ireland. [6] Immunobiology Laboratory, Department of Biomedicine, University Hospital Basel, Basel, Switzerland. [7] Institute of Functional Genomics, University of Regensburg, Regensburg, Germany. [8] Centre for Cancer Research, Hudson Institute of Medical Research, Clayton, VIC, Australia. [9] Cambridge Institute of Therapeutic Immunology & Infectious Disease, Department of Medicine, University of Cambridge, Cambridge, UK. [10] These authors contributed equally: Jennifer K Dowling, Remsha Afzal. ✉email: clairemccoy@rcsi.com

Macrophages are important immune cells that can exert either pro-inflammatory or anti-inflammatory functions, for innate as well as adaptive immune responses[1]. Metabolically, it has been shown that in vitro 'M1-like' inflammatory macrophages utilize aerobic glycolysis for the generation of ATP. This is accompanied with a downregulation of mitochondrial oxidative phosphorylation (OxPhos) and an accumulation of certain metabolites in the tricarboxylic acid (TCA) cycle, such as citrate and succinate[2,3]. Conversely, diverse anti-inflammatory stimuli including IL-10, IL-4, and IL-13 are 'M2-like' anti-inflammatory phenotype inducers. M2-like cells are shown to favour the use of OxPhos[4,5], which has been linked to specific alterations in macrophage mitochondrial dynamics[6–8].

IL-10 is an anti-inflammatory cytokine acting in an autocrine fashion in macrophages to limit inflammatory responses by decreasing the production of pro-inflammatory cytokines[9,10]. Simultaneously, IL-10 increases anti-inflammatory genes, typically in a STAT3 dependent manner[10–13]. Furthermore, inflammatory macrophages utilize arginine for the production of nitric oxide (NO), which IL-10 can limit by inhibiting the transcription of inducible nitric oxide synthase mRNA (Nos2)[14,15] or by enhancing the degradation of iNOS protein[10,16]. Concomitantly, IL-10 increases arginase expression to limit the availability of arginine for NO production[13,17]. In fact, IL-10 was shown to regulate macrophage glycolytic commitment by preserving OxPhos through its suppression of NO[18] or via suppression of mammalian target of rapamycin (mTOR)[19]. It has also been shown that IL-10, via STAT3, inhibits the pro-inflammatory microRNA miR-155[20]. IL-10 was shown to modulate miR-155 target genes suggesting a distinct mechanism that IL-10 uses to maintain an anti-inflammatory state in macrophages[20].

Here, we identify Arg2 as one of the most prominent metabolic genes regulated by the IL-10/miR-155 axis. We also show that IL-10-mediated induction of Arg2 protein is essential for skewing mitochondrial dynamics and bioenergetics in inflammatory macrophages towards an oxidative phenotype, particularly by enhancing activity of complex II (CII) at the electron transport chain (ETC). This work highlights Arg2 as a downstream mediator of IL-10 and provides a mechanism for its function as a resolver of inflammation.

## Results

**Arg2 is positively regulated by IL-10 in inflammatory macrophages.** We first identified the gene Arg2 when investigating regulation of inflammatory macrophages at the interface of the IL-10/miR-155 axis. A microarray was performed to examine gene expression changes between wild type (WT) murine bone marrow-derived macrophages (BMDM) stimulated with LPS (inflammatory) compared to those stimulated with LPS + IL-10 (anti-inflammatory) or to miR155$^{-/-}$ BMDM stimulated with LPS, miR-155$^{-/-}$(LPS) (predicted anti-inflammatory) (Fig. 1a and Supplementary Fig. 1a). Rotation gene set enrichment analysis (ROMER) indicated that the highest proportion of genes on this axis converged on metabolic pathways (Supplementary Fig. 1b). In particular, Arg2 displayed enhanced expression in both the WT (LPS + IL-10) and miR-155$^{-/-}$(LPS) groups (Fig. 1b), and was the only gene with a miR-155 binding site in its 3′UTR (Fig. 1b and Supplementary Fig. 1c)[21]. We confirmed Arg2 was a miR-155 target in macrophages by a luciferase assay, where Arg2 3′UTR activity was significantly suppressed in the presence of a miR-155 mimic, and reciprocally enhanced in the presence of the antagomir (Fig. 1c). Importantly, Arg2 protein levels were greater in miR-155$^{-/-}$(LPS + IL-10) compared to miR-155$^{+/+}$(LPS + IL-10), as opposed to Arg1 whose expression was consistent between genotypes (Fig. 1d).

Although IL-10 is broadly known to induce Arg1[10,13,22], only one inference suggested an IL-10 mediated increase of Arg2 in inflammatory macrophages by an affymetrix array[10]. Here, we significantly extend those studies by validating upregulation of Arg2 by exogenous IL-10 in LPS-treated murine and human macrophages compared to cells treated with LPS or IL-10 alone, at both the mRNA and protein level (Fig. 1e–h and Supplementary Fig. 1d). In contrast, Arg1 appears to be synergistically upregulated by LPS + IL-10 only in murine macrophages (Fig. 1h and Supplementary Fig. 1d). The dependency of autocrine IL-10 in Arg2 upregulation was confirmed when LPS mediated induction of Arg2 was reduced in BMDM treated with an IL-10Rα blocking antibody (Fig. 1i) and in Il10$^{-/-}$ BMDM stimulated with LPS (Fig. 1j). Moreover, the addition of exogenous IL-10 in LPS stimulated Il10$^{-/-}$ BMDM restored Arg2 expression (Fig. 1j). Additionally, in BMDM treated with STATTIC, an inhibitor of STAT3, significantly reduced levels of Arg2 mRNA and protein were observed (Supplementary Fig. 1e–g).

We next sought to investigate arginase catalytic activity (conversion of L-arginine to L-ornithine) by measuring the by-product urea in LPS, LPS + IL-10, and IL-10 stimulated BMDM. Firstly, in WT BMDM LPS treatment increased urea production but this was significantly enhanced by the addition of IL-10 in LPS + IL-10 treated cells (Fig. 1k), an effect that was lost in Il10$^{-/-}$ BMDM (Supplementary Fig. 1h). Pharmacological inhibition of the catalytic activity of arginase using the pan arginase inhibitor, nor-NOHA, confirmed the dependence of both isoforms on arginase activity in LPS + IL-10 stimulated cells (Supplementary Fig 1i, j). Furthermore, in order to establish the relative contribution of Arg1 and Arg2, we used siRNAs specific for each isoform (Supplementary Fig. 1k, l), and illustrated a reliance on both isoforms in the production of urea in LPS + IL-10 treated macrophages (Supplementary Fig. 1m). Arg1$^{-/-}$ mice die approximately 2 weeks after birth, hampering the capacity for BMDM harvesting[23]. However, we confirmed the contribution of Arg2 to arginine metabolism in Arg2$^{-/-}$ BMDM where there was a significant loss of urea production in Arg2$^{-/-}$(LPS + IL-10) cells (Fig. 1l). A concomitant increase in NO levels was also detected (Supplementary Fig. 1n), suggesting an absence of Arg2 may enhance the availability of L-arginine for iNOS.

**Arg2 is localized at the mitochondria.** In a further effort to identify independent roles for arginase isoforms, we focused on the fact that only Arg2 contains a mitochondrial targeting sequence (MTS)[24]. Although this infers constitutive localization at the mitochondria, this has not been thoroughly validated in macrophages[25] and translocation to the cytosol has been observed in endothelial cells[26], indicating that Arg2 localization is a dynamic process influenced by different conditions. Here, we definitively illustrate mitochondrial Arg2 localization in macrophages by employing a variety of techniques, and more specifically, assess the impact of IL-10 on this process. We firstly showed that Arg2 is localized to the mitochondria in comparison to Arg1, utilising an in vitro transcription/translation (IVTT) system (Fig. 2a). Overexpressing plasmids for Arg1 (pCMV-Arg1) and Arg2 (pCMV-Arg2) over a time-course of 60 min showed Arg2 is avidly imported into the mitochondria. However, this was not the case for Arg1 (Fig. 2a compare lanes 2–4 and 7–9). Furthermore, Arg2 remained imported following depolarization of the mitochondrial membrane via addition of FCCP (lane 10), which may indicate that Arg2 is localized in the inner mitochondrial membrane (IMM) as proteins targeted to the mitochondrial matrix require membrane potential to cross the IMM[27,28]. However,

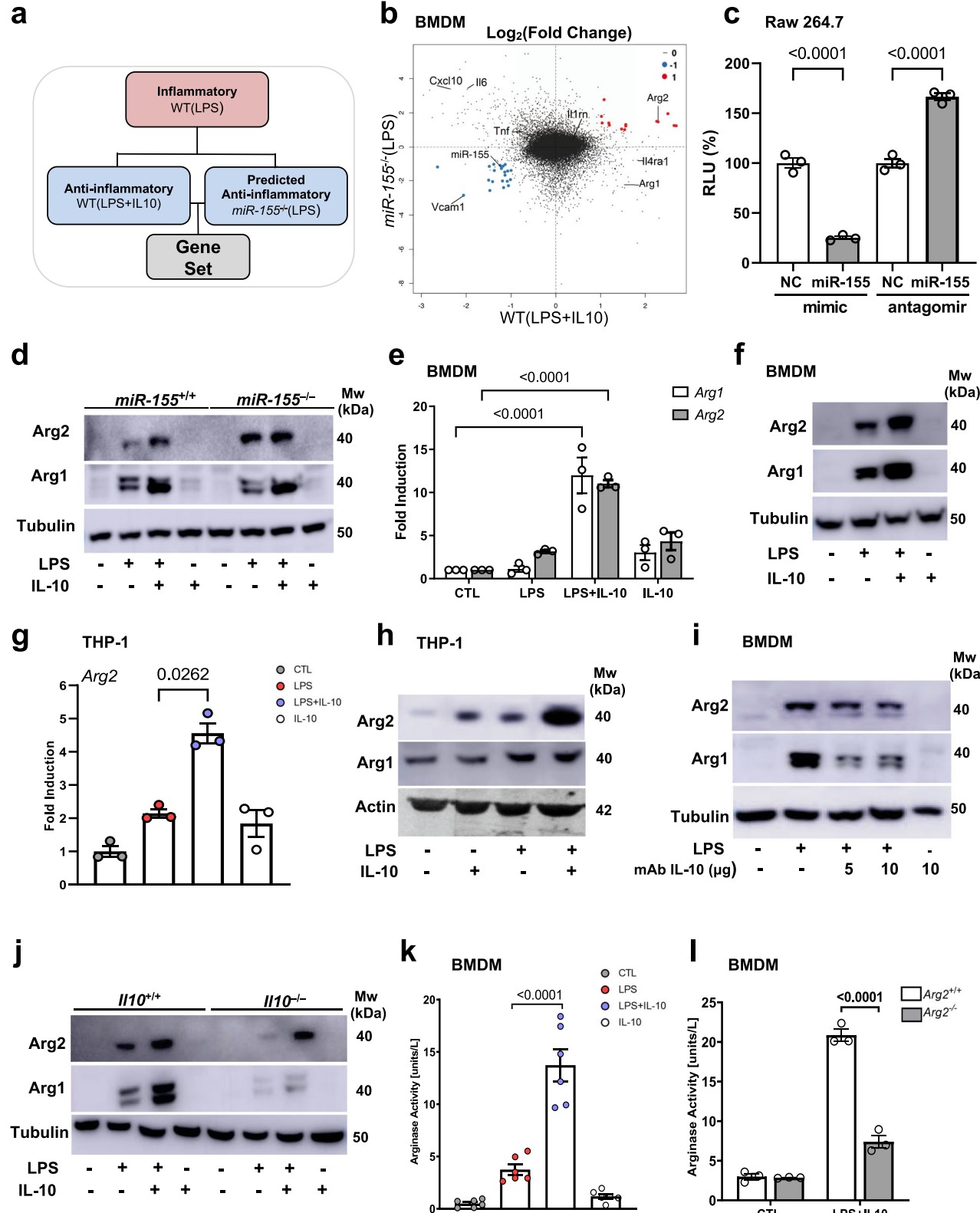

future studies will need to confirm the true nature of this localization.

We next wanted to ascertain if our stimulations induced translocation of Arg2 to or from the mitochondria (as previously reported for endothelial cells in response to oxLDL[26]). We confirmed with cellular fractionation in BMDM that Arg2 is localized and upregulated at the mitochondria in response to LPS

+ IL-10 in *Il10*[−/−] BMDM (Fig. 2b). This was not the case for Arg1, which was only detected in cytosolic fractions across treatments (Fig. 2b). Furthermore, using ImageStream[TM] flow cytometry, Arg2 was predominantly expressed and localized with the mitochondrial protein, Tom20, especially in LPS + IL-10 stimulated cells (Fig. 2c). Quantitative analysis revealed colocalization of Tom20/Arg2 in quiescent cells was 44.6%

**Fig. 1 Arginase-2 is regulated by an IL-10/miR-155 axis in inflammatory macrophages.** Macrophages (BMDM, Raw 264.7, or THP-1) were left untreated or stimulated with either LPS, LPS + IL-10, or IL-10 alone for 24 h. **a** Schematic of microarray design. Comparison of gene expression changes between WT (LPS) BMDM (inflammatory), WT(LPS + IL-10) BMDM (anti-inflammatory) or $miR$-$155^{-/-}$(LPS) BMDM, (predicted anti-inflammatory). **b** MA plot illustrating log fold-change between $miR$-$155^{-/-}$(LPS) and WT(LPS + IL-10) activated macrophages with common upregulated (red) and downregulated (blue) genes. Also shown are common inflammatory and anti-inflammatory genes in macrophages. **c** Luciferase activity assay of the $Arg2$ 3′UTR reported as relative light units (RLU) in the presence or absence of miR-155 mimic or inhibitor in Raw 264.7 cells ($n = 3$ independent experiments). Non-targeting control (NC) mimic or inhibitor are reported as reference and set at 100%. **d** Arg2 and Arg1 protein levels in $miR$-$155^{+/+}$ and $miR$-$155^{-/-}$ BMDM. Representative of $n = 3$ biological replicates. **e** RT-PCR analysis of $Arg2$ and $Arg1$ mRNA levels in BMDM ($n = 3$ biological replicates). Data compared by normalizing to $m18s$ and analyzing relative fold change to unstimulated controls. **f** Arg2 and Arg1 protein levels in BMDM. Representative of (at least) $n = 5$ biologically independent experiments. **g** RT-PCR analysis of $Arg2$ mRNA levels in THP-1 cells ($n = 3$ independent experiments. Data points represent average of technical triplicates per independent experiment). Data were compared by normalizing to $m18s$, and analyzing relative fold change compared to unstimulated controls. **h–j** Arg2 and Arg1 protein levels (representative of $n = 3$ biological independent experiments) in **h** THP-1 cells, **i** BMDM in the presence or absence of IL-10Ra (anti-IL-10R) antibody at indicated concentrations, **j** $Il10^{+/+}$ and $Il10^{-/-}$ BMDM. **k, l** Urea as output of arginase activity in: **k** wild-type BMDM ($n = 6$ biological replicates), and **l** $Arg2^{+/+}$ vs $Arg2^{-/-}$ BMDM ($n = 3$ biological replicates). **c, e, g, k, l** Data shown is mean with error bars representing ± SEM. Data were analysed by **c** Ordinary one-way ANOVA, and **e, l** two-way ANOVA, followed by **c, e** Tukey's and **l** Sidak's multiple comparisons post-hoc test. **g, k** Data for LPS vs LPS + IL-10 compared by paired two-tailed $t$-test. $P$-values are indicated on graphs for statistically significant comparisons.

(Supplementary Fig. 2a), which was boosted to 60% in LPS + IL-10 treated BMDM (Supplementary Fig. 2a). In addition, increased expression and localization of Arg2 at the mitochondria was also detected in Raw 264.7 macrophages treated with LPS + IL-10 (Supplementary Fig. 2b).

**Arg2 regulates mitochondrial dynamics.** Given the considerable expression of Arg2 at the mitochondria in LPS + IL-10 cells, we decided to examine mitochondrial morphology and dynamics under our experimental conditions. Mitochondrial function is intimately linked to their plastic structure which continuously changes according to the needs of the cell[29]. For example, LPS is a well-documented inducer of excessive mitochondrial fission[29–31], represented by small, punctate and fragmented mitochondria (≤1 μm). Contrarily, a state of 'fusion' in which mitochondria are elongated (≥3 μm) facilitates mitochondrial respiration and survival[32]. Assessing mitochondrial dynamics revealed LPS-only stimulated cells exhibited significantly increased mitochondrial fission compared to controls (≤1 μm approx. CTL:16%; LPS:85%) (Fig. 2d). Remarkably, however, this state of fission was not only significantly reduced in LPS + IL-10 (≤1 μm approx. 6%) but mitochondria were found predominantly in a state of fusion (≥3 μm approx. LPS:10%; LPS + IL-10:64%) (Fig. 2d). Next, we examined whether arginase enzymatic activity was essential for this morphological effect. We saw mitochondrial fusion was significantly decreased (≥3 μm approx. 22%) in LPS + IL-10 treated BMDM pre-treated with nor-NOHA. (Fig. 2e and Supplementary Fig. 2c). More importantly, we showed that $Arg2^{-/-}$ (LPS + IL-10) cells (≥3 μm approx. 18%) exhibited significantly lower levels of fused mitochondria compared to $Arg2^{+/+}$(LPS + IL-10) cells (≥3 μm approx. 64%) (Fig. 2f and Supplementary Fig. 2d).

**Arg2 increases mitochondrial respiration in macrophages.** IL-10 has been shown to influence mitochondrial features and associated functions such as mass, membrane potential ($\Delta\Psi_m$) and oxidative phosphorylation (OxPhos)[19]. Given the effect of Arg2 on mitochondrial dynamics, we investigated potential roles Arg2 may play in these IL-10 driven mitochondrial functions. We first demonstrated by flow cytometry that $Arg2^{-/-}$ BMDM were comparable in overall cellular size and intracellular complexity to $Arg2^{+/+}$ (Supplementary Fig. 2e, f). Furthermore, we found mitochondrial mass was unchanged between the genotypes in all experimental conditions (Supplementary Fig. 2g). However, we

observed that the membrane potential, $\Delta\Psi_m$, was significantly decreased in $Arg2^{-/-}$(LPS + IL-10) cells compared to $Arg2^{+/+}$(LPS + IL-10) (Supplementary Fig. 2h). As membrane potential is a driving force behind mitochondrial ATP synthesis, this finding led us to examine the effect of Arg2 on OxPhos. LPS-stimulated macrophages diminish their commitment to OxPhos as a means of energy production[33] while exogenous IL-10 can promote OxPhos in LPS-stimulated $Il10^{-/-}$ macrophages[19]. Certainly, we validated the ability of exogenous IL-10 to significantly restore OxPhos in $Il10^{-/-}$ BMDM treated with LPS, as evidenced by increased basal oxygen consumption rate (OCR), maximal respiratory capacity (MRC) and OxPhos-induced ATP production (Supplementary Fig. 3a, b). However, restoration of these parameters by IL-10 was significantly reduced in Raw 264.7 macrophages treated with siRNA against $Arg2$ (siArg2) compared to non-targeting controls (siNT) (Supplementary Fig. 3c). This finding was recapitulated in siRNA treated iBMDM (Fig. 3a, b and Supplementary Fig. 3d) and confirmed in $Arg2^{-/-}$(LPS + IL-10) BMDM (Fig. 3c, d).

Next we investigated the requirement of arginase catalytic activity in this process, and found that basal OCR and MRC were significantly blunted in $Il10^{-/-}$ (LPS + IL-10) cells pre-treated with nor-NOHA, with a concomitant reduction in OxPhos-induced ATP (Fig. 3e, f). Similar to BMDM, significantly downregulated OxPhos was observed in Raw 264.7 macrophages pre-treated with nor-NOHA (Supplementary Fig. 3e). To understand the relative contribution of Arg1 vs Arg2 on OxPhos, we looked at their in situ oxidative respiratory parameters in Raw 264.7 by overexpression plasmid studies, confirmed by immunoblot analysis (Supplementary Fig. 3f). Mitochondrial respiration results demonstrated that pCMV-Arg2 significantly boosted OxPhos (Fig. 3g), as shown by increased OCR parameters (Fig. 3h). However, no difference was detected between pCMV-Arg1 and the empty vector control, pCMV-EV (Figs. 3g and h). Finally, we specifically tested the importance of Arg2's catalytic activity by mutating its catalytic site at Histidine-145[34] to Phenylalanine (H145F). An arginase assay confirmed the catalytic inactivity of the H145F mutant (Fig. 3i), while immunoblotting illustrated comparable expression to wild-type pCMV-Arg2. (Supplementary Fig. 3g). Subsequently, we demonstrated that the catalytic site of Arg2 played a significant role in its regulation of OxPhos (Fig. 3j), as all parameters of Oxphos were downregulated in the H145F mutant compared to pCMV-Arg2 (Fig. 3k). Collectively, this is the first report of the Arg2 isoform influencing oxidative machinery in an immune cell type.

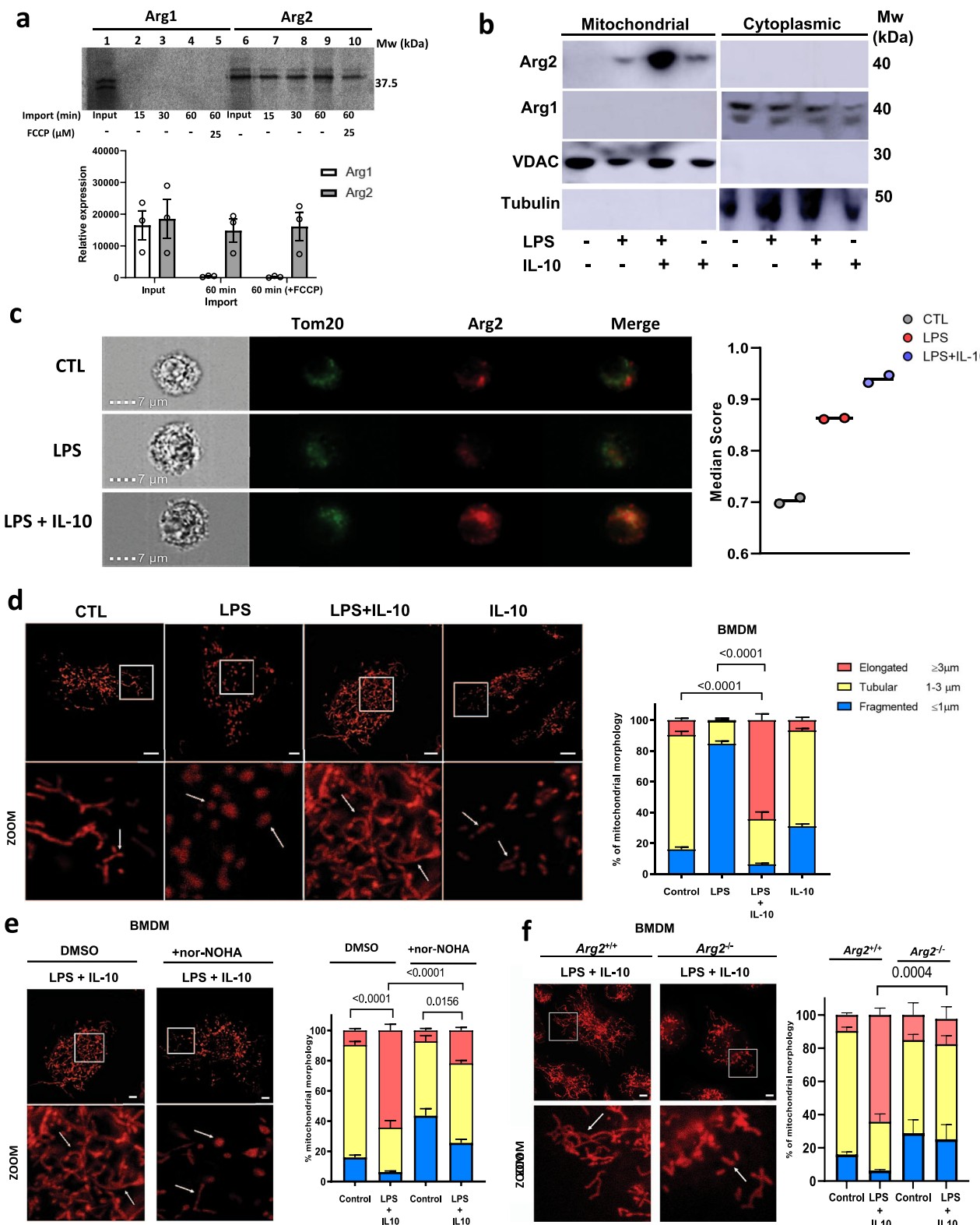

## Succinate dehydrogenase/complex II activity is increased by Arg2.

Having established that Arg2 boosts oxidative metabolism at the mitochondria, we wanted to unravel the details of this process and hence, looked towards the TCA cycle and ETC. Firstly we explored the impact of IL-10 on general TCA cycle metabolites by utilizing unbiased gas chromatography-tandem mass spectrometry (GC-MS). We observed that compared to controls, LPS + IL-10 mitochondrial wild-type BMDM pellets exhibited significantly increased levels of fumarate and malate (Supplementary Fig. 4a), two metabolites downstream of enzyme succinate dehydrogenase (SDH). This was corroborated by another study demonstrating significantly decreased levels of fumarate and malate in $Il10^{-/-}$ BMDM in response to LPS, compared to their wild-type counterparts[18].

SDH is a bi-functional enzyme that acts as a central component of the TCA cycle, and also serves as complex II (CII) at the ETC.

**Fig. 2 Arginase-2 regulates dynamics at mitochondria in response to IL-10 in inflammatory macrophages. a** IVTT assay for mitochondrial import of pCMV-Arg1 and pCMV-Arg2. Gels were exposed to phosphor-scene for detection of imported proteins. 1 µl of complete IVTT protein was loaded as input control (lanes 1 and 6). Time course over 60 min showing bands representing proteins bound to and found inside the mitochondrial membrane. FCCP is used as a depolarising control. Radiograph representative of three experiments ran and processed in parallel. Graph shows densitometry analysis of the pooled experiments by measuring relative expression. **b** Immunoblot of Arg1 and Arg2 in fractionated *Il10−/−* BMDM that were unstimulated (CTL), or stimulated with LPS, LPS + IL-10, or IL-10 alone. VDAC and β-tubulin are used as mitochondrial and cytoplasmic controls, respectively. Representative of three independent experiments. **c** ImageStream showing colocalization levels of Tom20 and Arg2 in BMDM that were unstimulated (CTL), or stimulated with LPS, or LPS + IL-10. (Left) Representative images shown from ~1000 events. Channels shown are bright field (Ch01), Tom20 (green/Ch02), Arg2 (red/Ch11), and a merge. Original magnification, ×60. (Right) Graph showing colocalization index of Tom20 and Arg2 in the different treatments over two independent biological experiments. Each data point represents median index of ~1000 processed images. **d–f** BMDM were either left unstimulated (CTL), or stimulated with LPS, LPS + IL-10 or IL-10 alone, and stained with MitoTracker Red-CMXRos. Mitochondrial morphology was observed by confocal microscopy. Lower panels show a higher magnification of the image within the white squares. Arrows show mitochondrial fusion and fission. Graphs show results of mitochondrial morphology analysis as a mean of $n = 3$ independent biological experiments. Scale bars represent 5 µm. **d** Wild-type BMDM, **e** BMDM pre-treated with DMSO or nor-NOHA and **f** *Arg2+/+* and *Arg2−/−* BMDM. **a, d–f** Data shown with error bars represents ± SEM. **d–f** Statistical significance was determined using two-way ANOVA with Tukey's post-hoc test for multiple comparisons. *P*-values are indicated on graphs where there is significance between elongated mitochondrial fractions.

To investigate the function of the ETC in a normal structural context, we examined individual complex activity by measuring the oxygen consumption rate (OCR) of permeabilized cells following treatment with Glutamate and Malate for complex I (CI), and Succinate for CII[35]. The OCR by CI and CII was significantly upregulated in LPS + IL-10 BMDM compared to unstimulated or LPS-stimulated cells (Fig. 4a). A possible role for IL-10 on complex activity was confirmed in *Il10−/−* BMDM where significantly reduced CI and CII activity was detected in resting *Il10−/−* cells compared to wild type controls (Supplementary Fig. 4b). Next, it was important to examine any impact of Arg2 in this setting. Interestingly, a significant decrease only in CII-specific OCR was detected in quiescent *Arg2−/−* compared to *Arg2+/+* BMDM, whereas CI specific OCR was comparable between genotypes (Fig. 4b). Furthermore, addition of exogenous IL-10 failed to boost CII-specific OCR in *Arg2−/−*(LPS + IL-10) cells comparatively (Fig. 4c). We also observed a significant gene dosage effect when comparing LPS + IL-10 treatments in Arg2 homozygous (*Arg2+/+*), heterozygous (*Arg2+/−*) and deficient (*Arg2−/−*) BMDM (Fig. 4d). Conversely, overexpression of Arg2 in Raw 264.7 macrophages boosted CII-dependent OCR (Fig. 4e). This was significantly blunted in the cells expressing the H145F catalytic mutant, demonstrating the importance of Arg2's catalytic activity in this process. Importantly, we confirmed that overexpression of Arg1 had no significant impact on CII activity relative to empty vector (pCMV-EV) control (Fig. 4e).

In an effort to fully validate our findings with respect to Arg2 and CII activity, we also measured the reduction of MTT to formazan, a common assay for SDH activity[36], which showed that *Arg2−/−* BMDMs had significantly lower levels of SDH activity in both resting and LPS + IL-10 treatment conditions (Fig. 4f). No difference was observed between *Arg2+/+* and *Arg2−/−* after LPS-only stimulation in this assay. The treatment of cells with the SDH inhibitor dimethyl malonate (DMM), as a positive control for the assay, reduced SDH activity in both genotypes to below baseline (Fig. 4f). Similar to our flux analyser experiments, overexpression of Arg2 in Raw 264.7 macrophages boosted CII activity in an MTT assay, and this was significantly blunted in H145F expressing cells (Supplementary Fig. 4c). Finally, we interrogated CII specific activity using a single wavelength spectrophotometer technique and expressing it as a ratio to citrate synthase (CS) specific activity in order to normalize activities to mitochondrial mass, as previously described[37]. Results demonstrated that CII specific activity was significantly decreased in resting *Arg2−/−* BMDM compared to *Arg2+/+* controls (Fig. 4g). Additionally, exogenous IL-10 could significantly restore CII specific activity in *Arg2+/+* (LPS +

IL-10) cells when compared to *Arg2+/+* (LPS). However, this ability of IL-10 to restore CII specific activity was lost in *Arg2−/−* (LPS + IL-10) BMDM (Fig. 4g). These results implicating a role for Arg2 in CII specific activity were upheld when measuring cellular succinate and fumarate in LPS + IL-10 *Arg2−/−* BMDM compared to wild-type controls. Here we demonstrated that IL-10 reduced overall succinate (Fig. 4h) and boosted fumarate levels in LPS + IL-10 stimulated BMDM compared to LPS alone (Fig. 4i). This effect was once again lost in *Arg2−/−*(LPS + IL-10) BMDM (Fig. 4h-i).

**Arg2 regulates HIF-1α and IL-1β during acute inflammation**. The mechanism of LPS-stimulated breaks in the TCA cycle and abnormal ETC activity are broadly considered inflammatory[38]. In such instances, studies have reported an increased inflammatory phenotype due to direct generation of mitochondrial ROS (mtROS). This has been intimately linked with the stabilisation of HIF-1α, increased production of cytokine IL-1β, and a reduced anti-oxidant response[2,39]. We detected a significant increase in mtROS in *Arg2−/−*(LPS + IL-10) BMDM compared to *Arg2+/+*(LPS + IL-10) (Fig. 4j) along with enhanced protein levels of HIF-1α in *Arg2−/−*(Fig. 4k). Importantly, examination of IL-1β levels under our conditions revealed that the suppressive effect of IL-10 on IL-1β in *Arg2+/+*(LPS + IL-10) cells was completely lost in *Arg2−/−*(LPS + IL-10) (Fig. 4l). Notably, there were no significant differences in TNF or IL-6 levels between *Arg2+/+*(LPS + IL-10) and *Arg2−/−*(LPS + IL-10) (Supplementary Fig. 4d). Furthermore, reduced levels of Nrf2, a transcription factor known to induce expression of anti-oxidant genes, was detected in *Arg2−/−* cells across treatments (Supplementary Fig. 4e).

These findings suggested that IL-10 works through Arg2 at the mitochondria to mediate its anti-inflammatory effects on IL-1β production. Next, we examined whether Arg2 is limited in situations where IL-10 is compromised by performing an in vivo LPS challenge in *Il10+/+* and *Il10−/−* mice. As expected, TNF, IL-6 and IL-1β protein levels were significantly increased in the peritoneal lavage of *Il10−/−* (Supplementary Fig. 4f). Interestingly, *Arg2* mRNA levels were reduced significantly in *Il10−/−* PECs (Supplementary Fig. 4g) and there was an indicated absence of Arg2 protein in tissue taken from the spleen in *Il10−/−* with no protein expression changes apparent in Arg1 (Supplementary Fig. 4h). Finally, we examined responses after an in vivo LPS challenge in *Arg2+/+* and *Arg2−/−* mice. We observed that comparable to our ex vivo results in primary BMDM, there was significantly enhanced HIF-1α expression in spleens of *Arg2−/−* mice (Fig. 4m). Furthermore, even though there was a significant

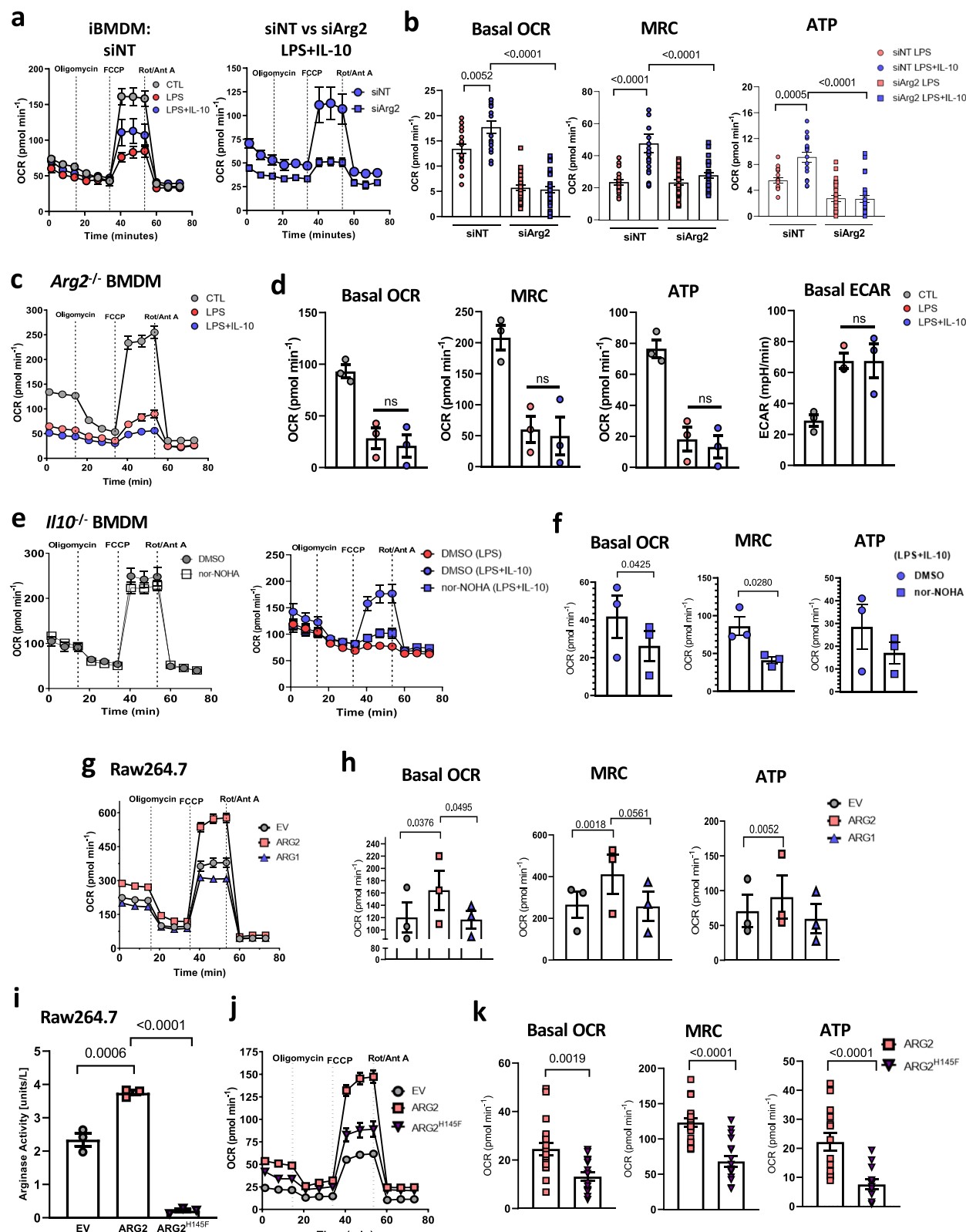

increase in IL-10 secretion in $Arg2^{-/-}$ mice in the peritoneal lavage, IL-1β secretion levels remained significantly higher in these mice compared to wild-type mice (Fig. 4n). Interestingly, this result was mirrored in vitro following siRNA knockdown of $Arg2$ in LPS + IL-10 treated iBMDM (Supplementary Fig. 4i). These results collectively indicate that the suppressive effect of IL-10 on IL-1β is lost in the absence of Arg2.

## Discussion

Here, we show that IL-10 mediated upregulation of Arg2 is essential for skewing mitochondrial dynamics and bioenergetics in inflammatory macrophages towards an anti-inflammatory, oxidative phenotype. This work is supported by previous experiments conducted in H.pylori mediated gastritis where macrophages from Arg2 deficient mice presented with a more

**Fig. 3 Arginase-2 enhances oxidative phosphorylation in inflammatory macrophages. a–h, j, k** Oxygen consumption rates (OCR) were assessed by real-time metabolic flux assay by addition of Oligomycin (1 μM), FCCP (0.9 μM), and Rotenone + Antimycin A (Rot/Ant A) (0.5 μM) sequentially. **a** (Left) unstimulated siNT and si*Arg2* immortalized BMDM (iBMDM) cells and (right) LPS and/or LPS + IL-10 treated siNT vs si*Arg2* iBMDM (representative of $n = 2$ biological experiments). **b, d, f, h, k** Quantitative changes for the basal oxygen consumption rate (basal OCR), maximal respiratory capacity (MRC), and Oxphos-induced ATP levels. **b** LPS and LPS + IL-10 treated iBMDM cells that were either transfected with siNT or si*Arg2*. Data points shown are pooled 15–20 technical replicates of two independent experiments. **c** *Arg2*$^{-/-}$ BMDM (trace representative of $n = 3$ biological repeats). **d** Quantitative changes in oxidative parameters for unstimulated, or LPS/ LPS + IL-10 stimulated *Arg2*$^{-/-}$ BMDM ($n = 3$ biological experiments). **e** *Il10*$^{-/-}$ BMDM (left) unstimulated, or (right) LPS/LPS + IL-10 stimulated ($n = 3$ biological experiments). Cells were either pre-treated with DMSO or with 150 μM nor-NOHA for 1 h before addition of IL-10 and/or LPS stimulations. **f** Quantitative changes in *Il10*$^{-/-}$ BMDMs pre-treated with DMSO or nor-NOHA in oxidative parameters for LPS + IL-10 treated cells. Data points are biological replicates ($n = 3$). **g** Raw264.7 macrophage expressing pCMV-EV, pCMV-Arg1 or pCMV-Arg2. Trace representative of three independent experiments. **h** Quantitative oxidative parameters changes in Raw264.7 cells transfected with overexpression plasmids from (**g**). Data points are three independent experiments ($n = 3$). **i–k** Raw 264.7 macrophage expressing pCMV-EV, pCMV-Arg2 or catalytic dead mutant pCMV-H145F. **i** Arginase activity assay. Data points indicate three independent experiments. **j** Mitochondrial OCR trace representative of two independent experiments. **k** Arg2 vs H145F OCR parameters. Data points show pooled 15–20 technical replicates from two independent experiments. **a–k** Data is presented for all traces and scatter plots as mean ± SEM. **d, f, h** Each biological or independent data point shown had 10–20 technical replicates. Data were analysed for statistical significance using a two-tailed Student's *t*-test, and **b, l, k** using ordinary one-way ANOVA with Tukey's post-hoc test for multiple comparisons. Respective *p*-values are indicated on graphs for results that were statistically significant.

inflammatory phenotype, including increased NO and pro-inflammatory cytokine milieu[40]. We show that IL-10's enhancement of Arg2 favours mitochondria in a state of fusion. In support of this finding, a positive correlation between systemic IL-10 levels in 175 healthy human patients and genes involved in mitochondrial fusion was previously reported[41]. Certainly, mitochondrial dynamics, specifically fusion, has been shown to influence OxPhos[42,43]. This is linked with our other main finding that Arg2 facilitates oxidative respiration via its impact on CII activity at the ETC. Interestingly we found these effects to be, at least in part, dependent on Arg2's catalytic activity, suggesting ornithine production may play an important role in this process. Previous work by others has highlighted that IL-4 upregulation of ornithine is essential for hypusination of the eukaryotic translation initiation factor 5a (eIf5a), which works to maintain integrity of mitochondrial TCA cycle and OxPhos proteins[44]. Whether Arg2's impact on mitochondrial respiration is dependent on the production of ornithine requires further investigation.

As evidenced by global metabolomics studies, macrophages fluctuate their TCA cycle metabolites in response to inflammatory stimuli[45]. Specifically, LPS-induced inflammatory macrophages increase their succinate levels with decreased alpha-ketoglutarate levels. Work from Baseler et al.[18] demonstrated that the LPS-induced accumulation in succinate in wild-type BMDM was abolished in *Il10*$^{-/-}$ BMDM, yet the reductions of alpha-ketoglutarate, indicative of the LPS-induced "TCA break", remained completely intact[18]. This suggests that autocrine IL-10 acts to relieve the TCA cycle "break" by increasing metabolites downstream of succinate, such as fumarate and malate. In that regard, SDH, a central enzyme of the TCA cycle, converts succinate to fumarate, and is shown to be inhibited in LPS-stimulated macrophages[39,46]. Adding to these studies, we have shown that IL-10 decreases succinate, enhances SDH/CII activity and increases fumarate production in inflammatory macrophages, an effect we demonstrate to be dependent on Arg2. Moreover, we show that in *Arg2*$^{-/-}$, HIF-1α and IL-1β expression is elevated. This is interesting considering excess succinate can lead to HIF-1α stabilization[2], while patients harbouring mutations in SDH have increased HIF-1α activity[47,48] and circulating succinate levels[49]. Collectively, our work demonstrates that Arg2 is integral in resolving the inflammatory state of the cell.

The fumarate analogue dimethyl fumarate, DMF, is an immunomodulatory drug used for the treatment of inflammatory disorders like psoriasis and multiple sclerosis (MS). Furthermore, monocytes from DMF-treated relapsing-remitting MS patients

have decreased expression of the pro-inflammatory miR-155[50]. DMF has also been shown to reduce pro-inflammatory cytokines such as IL-1β[51,52], enhance mitochondrial oxidative respiration[53] and boost the anti-oxidant Nrf2 pathway[54]. In our studies, we show a similar profile with Arg2, where we observed enhanced Arg2 in *miR-155*$^{-/-}$ macrophages. Furthermore, a loss of Arg2 in LPS-stimulated macrophages resulted in increased mtROS, HIF-1α and IL-1β secretion, along with reduced Nrf2 expression, despite the presence of exogenous IL-10. Certainly, investigating metabolic processes has led to the discovery of therapeutic metabolites[55] and a better understanding of the mechanisms of action for current immunomodulatory drugs. Considering IL-10 has had a checkered history as a therapeutic agent, one can envision that modulation of its targets, like Arg2, could provide an alternative route for the treatment of inflammatory disorders. We currently await further analysis of this metabolic enzyme as a regulator of immunity.

Limitations of the study: Since its discovery, studies have postulated the likelihood of alternative functions for Arg2, the lesser studied arginase isoform. It is intriguing that high expression of the Arg2 isoform at the mitochondria, can facilitate an increase in SDH activity to impact on the inflammatory status of the cell. Future studies should focus on the potential interactions between SDH and Arg2 at the macrophage's mitochondria, and more broadly in other cell types.

## Methods

**Mice.** All mice were bred and housed in on-site barrier-controlled facilities having a 12-h/12-h light/dark cycle with ad libitum access to food and water. Breeding was approved by the Ethics Committee of the Royal College of Surgeons in Ireland (REC-842), under license from the Ireland Health Products Regulatory Authority (AE19127/001) and conformed to the Directive 2010/63/EU of the European Parliament. *miR155*$^{-/-}$ C57BL6/J, *Arg2*$^{-/-}$ C57BL6/J and wild-type (WT) C57BL6/J mice were all purchased from the Jackson Laboratories. *Il10*$^{-/-}$ mice were kindly provided by Ed Lavelle (Trinity College Dublin, Dublin, Ireland). *Il10*$^{-/-}$ mice were originally from the Jackson Laboratories (Bar Harbor, Maine) and bred in-house.

**LPS challenge in vivo.** All mice were on a C57BL/6J background, with wild-type mice obtained from the Bioresources Unit, Trinity Biomedical Sciences Institute. *Il10*$^{-/-}$ mice were housed in the Bioresources Unit, Trinity Biomedical Sciences Institute in Trinity College Dublin (TCD). Wild type and *Arg2*$^{-/-}$ littermate mice were housed in the BRF unit at the Royal College of Surgeons in Ireland. Mice were used at 8-12 weeks of age. Animals were maintained according to the regulations of the Health Products Regulatory Authority (HPRA). Animal studies were approved by the TCD and RCSI Animal Research Ethics Committee (Ethical Approval Number 091210) and were performed under the appropriate licence (Licence Number AE19136/P079 for *Il10*$^{-/-}$ mice and A19127-P045 for *Arg2*$^{-/-}$ mice). Age matched female mice were injected by the intraperitoneal (i.p.) route with 10 mg/kg of LPS (E. coli 0111:B4, Invivogen) and culled after 8 h. To detect

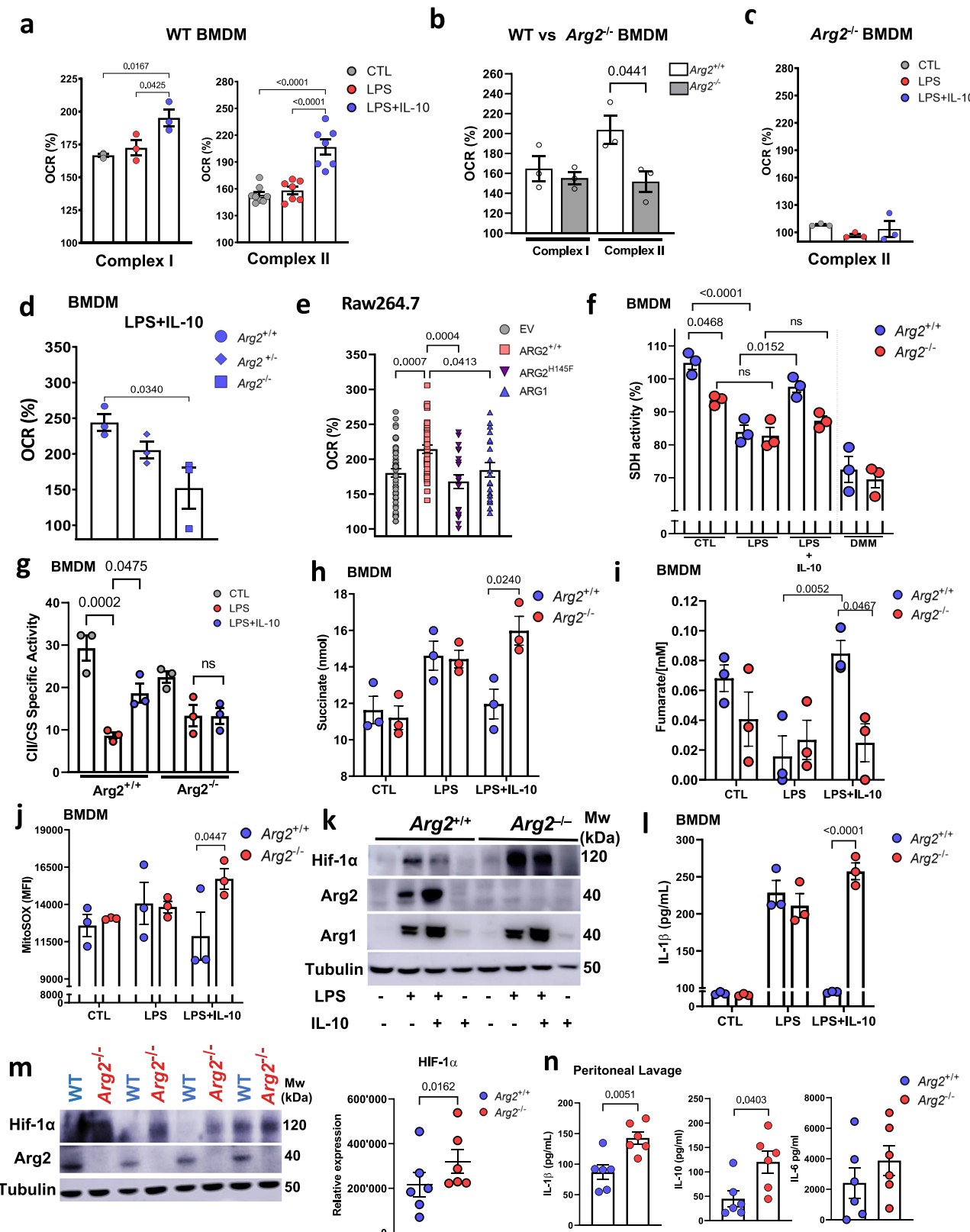

cytokines, peritoneal fluid and sera were collected 8 h following i.p. injection of LPS and stored at −80 °C. IL-1β was quantified by ELISA (R&D Systems). Levels of TNF, IL-6, IFNγ were calculated by cytokine bead array using the mouse inflammation kit (BD Biosciences). Peritoneal cells (PECs) were isolated by flushing the peritoneum cavity with PBS containing 5 mM EDTA. Cells were centrifuged and total RNA isolated using the RNeasy Plus Mini kit (Qiagen) and stored at −80 °C. Spleens were excised, cut in half, snap-frozen in liquid nitrogen, and stored at

−80 °C until time of assay. RNA was extracted from one half of the spleen by TRIzol extraction and the second half homogenised, assayed for protein quantification by BCA assay (Pierce) and western blotting. The yield and purity of RNA from spleen and PEC samples was determined using a Nano-Drop 1000 Spectrophotometer (VICTOR3000). RNA was converted to cDNA using a reverse transcription kit (Applied Biosystems). mRNA expression of *Tnf*, *Il1b*, *Arg2*, *Inos* and *Il6* was determined by Real-time quantitative PCR (RT-PCR) using SYBR

**Fig. 4 Arginase-2 influences complex-II activity and regulates IL-1β secretion. a–e** XFe96 based comparison of complex I and/or complex II specific OCR in untreated (CTL) BMDM or stimulated cells (with LPS or LPS + IL-10 for 24 h). Data plotted as oxygen consumption rate percentage (OCR%) increase after addition of substrate: **a** wild-type cells. $n = 3$ (complex I) and $n = 7$ (complex II) biological replicates. **b** Quiescent wild-type ($Arg2^{+/+}$) vs $Arg2^{-/-}$, **c** CTL vs stimulated $Arg2^{-/-}$, **d** LPS + IL-10 stimulated $Arg2^{+/+}$, $Arg2^{+/-}$, and $Arg2^{-/-}$; **b–d** Data shown are biological $n = 3$. **e** Complex II specific OCR in Raw 264.7 overexpressing pCMV-SPORT6 plasmids for EV (empty vector), Arg2, H145F, and Arg1. Data show 20 technical replicates (for H145F and Arg1) and 40 technical replicates (for EV and Arg2) over two independent experiments. **f** SDH activity as determined by MTT assay in $Arg2^{+/+}$ and $Arg2^{-/-}$ BMDM. SDH inhibitor dimethylmalonate (DMM) used as a control ($n = 3$ biological replicates). **g** Complex II specific activity measured by UV-VIS spectrophotometry in lysates from $Arg2^{+/+}$ and $Arg2^{-/-}$ BMDM. Activity was normalized to citrate synthase (CS) as control ($n = 3$ biological triplicates), **h–l** Comparison of wild-type ($Arg2^{+/+}$) and $Arg2^{-/-}$ BMDM for levels of **h** succinate, **i** fumarate, **j** mitochondrial ROS (mtROS) by staining with mitoSOX, **k** protein expression for Hif-1α, Arg2, Arg1, and β-tubulin (as loading control); **l** IL-1β ELISA comparing secretion levels in supernatants. **h–j, l** $n = 3$ biological replicates, and **k** representative of $n = 3$ independent experiments; **m, n** $Arg2^{+/+}$ and $Arg2^{-/-}$ C57Bl/6J mice were given i.p injection of LPS at 10 mg/kg for 8 h: **m** (Left) Hif-1α, Arg2, and β-tubulin in spleen. Blot representative of six mice each from wild-type (WT) and $Arg2^{-/-}$. (Right) densitometry analysis showing relative expression levels of Hif-1α ($n = 6$); **n** IL-1β, IL-10, and IL-6 secretion levels in peritoneal lavage ($n = 6$ biological replicates). **a–j, l–n** Data are presented for scatter plots as mean ± SEM. Data were analysed by **a–g, l** ordinary one-way ANOVA with Tukey's multiple comparisons, **h, j, n** two-tailed t-test with Welch's correction, and **l, m** two-tailed paired t-test. Wherever significant, exact p-value is given on the figure.

Green probes on a 7900 HT Real-Time PCR System (Applied Biosystems). Fold changes in expression were calculated as described below.

**Bone marrow-derived macrophage generation**. Bone marrow was isolated from 6 to 12 weeks old adult female littermates. Mice were euthanized in a $CO_2$ chamber and death was confirmed by cervical dislocation. Femurs and tibias were isolated in sterile conditions, and the bone marrow was flushed out using Dulbecco's Phosphate Buffered Saline (DPBS). Marrow was spun and incubated with red blood cell (RBC) lysis buffer (Sigma) to remove red blood cells. A single cell suspension was prepared by passing the cells through a 70 μm cell strainer (Corning). They were then plated in 10 cm petri dishes in complete DMEM (Dulbecco's Modified Eagle Medium (Sigma-D5796) supplemented with 10% heat-inactivated Fetal Bovine Serum (FBS) (Sigma-F9665) and 1% Penicillin/Streptomycin (100 U/ml) (Sigma). 20% L929 cell supernatant was also added to the culture to induce BMDM differentiation via MCSF, after which cells were incubated for 6 days at 37 °C and 5% $CO_2$ levels. Cells were counted and tested for viability using Trypan Blue staining (Invitrogen) and a haemocytometer. They were then re-plated for experiments in complete DMEM supplemented with 10% L929 cell supernatant. Unless stated otherwise, $5 \times 10^5$ cells/ml were used for ex vivo experiments.

**Cultured cell lines**. Raw264.7, L929, and THP-1 cultured cell lines were obtained from American Type Culture Collection (ATCC) and cultured as suggested by the supplier. The immortalised BMDM (iBMDM) were a kind gift from Prof Luke O' Neill, Trinity College Dublin, as used in previous studies[56,57]. Cells were routinely tested to be Mycoplasma negative. They were incubated in 37 °C with 5% $CO_2$ levels. Cell viability was determined using Trypan Blue and counted with a haemocytometer. Raw 264.7 and iBMDM cells were passaged every second and fifth day (1:5), respectively in T75 flasks. They were cultured in complete DMEM (supplemented with 10% heat-inactivated FBS and 1% penicillin/streptomycin). All experiments were carried out in early passage numbers, with passage number not exceeding 11 at most. Unless stated otherwise, Raw264.7 cells were plated at a density of $1 \times 10^6$ cells/ml and iBMDM were plated at a density of $5 \times 10^5$ cells/ml for in vitro experiments. THP-1 cells were cultured in complete RPMI 1640 (Sigma) supplemented with 2 mM L-glutamine, 10% FBS, and 1% penicillin/ streptomycin. They were plated at a density of $2.5 \times 10^5$ cells/ml and differentiated using 10 ng/ml phorbol-12-myristate-13-acetate (PMA) (Sigma) for 7 h, after which the media was replaced with PMA-free medium.

L929 cells were passaged every second day (1:5) in T175 flasks. They were cultured in RPMI-1640 containing 10% heat-inactivated FBS and 1% penicillin-streptomycin. To generate L-cell supernatant for BMDM differentiation, $20 \times 10^6$ cells were plated in 40 ml of complete RPMI-1640 in T175 flasks for 10 days after which the media was filtered and frozen at −20 °C until use.

**Stimulations**. Fresh media was added to the cells before stimulation experiments. Recombinant mouse IL-10 (R&D Systems) from a 1 mg/ml stock was diluted to a 20 μg/ml working stock in 0.1% BSA solution. It was then added to cells at a final concentration of 20 ng/ml (and 100 ng/ml for metabolic studies). LPS (Sigma E.coli O111:B4) was diluted from stock concentration of 1 mg/ml in complete DMEM, and used at a final concentration of 100 ng/ml. For LPS + IL-10 treatments, cells were pre-incubated for five minutes with IL-10 prior to the addition of LPS. Cells were typically stimulated for 24 hours before conducting further assays. Inhibitor experiments involved pre-treatment before LPS/IL-10 addition. Cells requiring incubation with inhibitor nor-NOHA (150 μM) (Cayman Chemicals) were pre-treated 1 h prior to stimulation. Dimethyl Sulfoxide (DMSO) and/or DPBS (Sigma) were used as controls for inhibitor experiments. For THP-1 cells, 100 ng/ml human recombinant IL-10 (R&D Systems) was used.

**Affymetrix array**. Primary BMDM from wild-type (WT) and miR-155 knockout animals (KO) were stimulated in biological duplicates with LPS (100 ng/ml) or LPS + IL-10 (20 ng/ml) to generate four groups as follows: WT.LPS ($n = 2$), WT.LPS. IL-10 ($n = 2$), KO.LPS ($n = 2$) and KO.LPS.IL10 ($n = 2$). RNA was extracted using the RNeasy mini kit (Qiagen) and sent to Almac Diagnostics for Affymetrix microarray analysis. Spectrophotometer QC was performed to assess the concentration and purity of the samples and Bioanalyzer QC was performed to assess the integrity of the samples. All eight samples were found to meet the QC requirements for downstream array processing. All samples were amplified using the NuGEN's Ovation RNA Amplification System V2 with the Ovation WB Reagent. Following amplification, all generated cDNA samples were quality controlled using both the spectrophotometer and the Agilent 2100 Bioanalyzer. All samples passed the quality assessment. In total eight samples proceeded to be fragmented and labelled using the NuGEN Encore Biotin module, before being hybridised onto Mouse Genome 430_2 array in accordance with the NuGEN guidelines for hybridisation onto Affymetrix GeneChip® arrays. Array image analysis performed on CEL files confirmed no surface or background artefacts. No outliers were identified as examined by measuring parameters such as background intensity, Raw noise (Q) value, scaling factor and GAPDH/Actin ratio.

Bioinformatic Analysis raw data was RMA normalised using the affy package[58]. Data quality was assessed by box plots of expression values before and after normalisation. Differential expression analysis was performed using the limma package[59]. Samples were assigned to treatment groups and a linear model fit using the lmFit function. Mouse donor was incorporated as a blocking factor in the linear model, inter-subject correlation obtained using the duplicate Correlation function[60]. Y-linked genes and Xist were excluded from differential expression analyses. Moderated t-statistics were calculated for WT.LPS.IL10 versus WT.LPS and KO.LPS versus WT.LPS using the eBayes function[61]. Differentially expressed probes were selected using an unadjusted p-value < 0.05 and an absolute $\log_2$(fold change) >1. Gene set enrichment analysis was performed using the romer function[59,62] using the C5 Gene Ontology (GO) gene sets[63,64] from the Molecular Signatures Database[65].

**Metabolomics**. $5 \times 10^6$ BMDM were plated in complete DMEM in T-25 flasks (25 cm²). Cells were left untreated, treated with LPS (100 ng/ml) in the presence or absence of IL-10 (20 ng/ml), or IL-10 alone for 20 h. Cells were scraped in 600 μL 80% (v/v) MeOH and cell pellets frozen at −80 °C for 48 h. For extraction cells were pelleted, washed twice with 80% methanol. Extract was dried in a speed-dry evaporator and afterwards stored at −80 °C till processing. Intracellular TCA cycle intermediates were quantified using GC-MS. Stable isotope labelled standards were added to the cell lysates, which were dried (CombiDancer Hettich AG) and subjected to methoximation and silylation. Then, 1 μl of the derivatized sample was injected using splitless injection mode. Gas chromatography–mass spectrometry (GC-MS) data were analysed using the software Mass Hunter (version B.07.01/ Build 7.1.524.0). Quantification was achieved by individual calibration curves based on the analyte to internal standard area ratio[66].

**Mitochondrial isolation**. BMDM from bacterial 10 cm dishes were harvested, washed in DPBS, and resuspended in 3 ml of Solution A (20 mM HEPES pH 7.6, 220 mM mannitol, 70 mM sucrose, 1 mM EDTA, 0.5 mM PMSF, and 2 mg/ml BSA) (Sigma). Cells were left on ice for 15 min in Solution A and then homogenized in a glass homogenizer with a Teflon pestle, giving the cells about 60 strokes. Homogenates were then centrifuged at $800 \times g$ for 10 min at 4 °C to pellet non-lysed cells, cell debris, and nuclei. The supernatant was retained and centrifuged at $12,000 \times g$ for 20 min at 4 °C for 15 min. The supernatant was then removed and the pellet was resuspended in ~0.8 ml solution B (Solution A without BSA) and recentrifuged at the same speed. The step was repeated twice more and

the final pellet was resuspended in 100 μl solution B. The amount of mitochondrial protein was quantified using a BCA assay.

**Mitochondrial Import assays**. In vitro transcription/translation (IVTT) reactions were performed using an RNA polymerase with SP6 promoter transcriptional compatibility to our pCMV plasmids. pCMV-Arg1 or pCMV-Arg2 were used as templates in separate IVTT reactions with S35 Met and Cys to generate radio labelled protein. Mitochondria were isolated and IVTT protein added to mitochondrial extracts in import buffer and incubated for 60 min at 37 °C. Samples were then were run on a gel, including a lane for complete IVTT protein as input control (In). Gels were exposed to phosphor-scene for detection of imported proteins. Samples were treated with 25 μM mesoxalonitrile 4-trifluoromethoxyphenylhydrazone (FCCP) to depolarise mitochondrial membranes over a time course of 60 min. After the import assay was done, an SDS-PAGE gel was run, which was dried for imaging with a phosphoimager.

**siRNA-induced knockdown**. Accell® siRNA against *Arg2* was obtained from Dharmacon (GE Healthcare; EQ-040721-00-0002) in dried-down format (2 nmol/siRNA: set of 4 siRNAs). Other reagents from Dharmacon (CK-005000-R1-02) included Accell siRNA delivery media and Accell non-targeting control kit siRNA (5 nmol/siRNA; set of 4 siRNAs). For siRNA stock preparation, 5× Accell siRNA Buffer (provided with the Accell non-targeting control kit) was diluted to 1× siRNA Buffer using RNAase-free water under sterile conditions. 100 μM of stock siRNA solutions were prepared using 1X siRNA Buffer. The stock solutions were placed on a shaker for 60 minutes at 37 °C when reconstituted for the first time. For experiments, $4 \times 10^5$ Raw 264.7 or $2.5 \times 10^5$ iBMDM cells were plated per well in a 24-well plate in complete DMEM and left to propagate overnight in the incubator. The next day media was removed from the cells and they were transfected using newly reconstituted siRNA at 1 μM diluted in fresh Accell delivery media that was supplemented with 2.5% heat-inactivated FBS. The cells were incubated at 5% CO₂ and 37 °C for the next 48 h, after which they were washed, collected, and counted for number and viability using Trypan Blue (Invitrogen). This allowed only viable cells to be plated at the appropriate cell numbers for another 24–48 h in fresh complete DMEM, and were then treated according to the assay being conducted.

**pCMV-SPORT6 transient transfection**. $1.5 \times 10^6$ Raw-264.7 cells were seeded in a 6-well plate in complete DMEM and left overnight to propagate. The following day, fresh serum-free DMEM (without antibiotic) was added to the cells for transfection. Overexpression plasmids were purchased as pCMV-SPORT-6-Arg2 and pCMV-SPORT6-Arg1 (Invitrogen). They were transfected at 500 ng/well in the respective wells using Lipofectamine 3000 (Invitrogen). For control wells, an empty vector pCMV-SPORT6 plasmid was used. Cells were incubated at 37 °C and 5% CO₂ for 4 h with the transfection mix in serum and antibiotic free media, after which media was replaced with complete DMEM. Cells were incubated for a further 24 h for Immunoblotting. For metabolic profiling assays, cells were scraped 24 hours post-transfection, counted for number and viability and transferred to XFₑ96-well plates (Agilent) for another 24 h before conducting the assay.

**Immunoblotting**. Cells were resuspended in low-stringency lysis buffer (50 mM HEPES (pH 7.5), 100 mM NaCl, 10% glycerol (v/v), 0.5% Nonidet P-40 (v/v), 1 mM EDTA, 1 mM sodium orthovanadate, 0.1 mM PMSF, 1 mg/ml aprotinin, and 1 mg/ml leupeptin) (Sigma). The resulting suspension was centrifuged at $12,000 \times g$ for 20 min at 4 °C, and supernatants were collected and used for SDS-PAGE. Protein samples were normalised by BCA protein assay (Pierce), and denatured by the addition of 5× SDS sample buffer containing 0.2 M DTT and heated at 95 °C for 10 min. 20 μg sample protein was loaded per well and samples were resolved on 4–12% gradient Bis-Tris gels (Invitrogen). Transfer was done for 60 min onto PVDF membranes using mini-blot modules as per the manufacturer's instructions (Invitrogen). Membranes were blocked in 5% (w/v) dried milk in Tris-buffered saline-Tween (0.05% v/v) (TBST) for 1 h at room temperature. Membranes were then incubated with primary antibody (concentrations mentioned in Supplementary Table 1) diluted in 5% milk/TBST overnight at 4 °C, followed by incubation in the appropriate HRP-conjugated secondary antibody for 1.5 h at room temperature. Blots were washed in TBST (4 × 10 min), incubated in LumiGlo™ ECL Reagent (Cell Signaling Technology) for 1 min, and imaged by chemiluminescence using the Amersham imager. Quantification of the blots was performed using ImageStudio Lite. Briefly, individual bands were quantified for relative intensity of signal, and then normalized to loading control protein signal (e.g., β-Tubulin). Uncropped blots are presented in the Source Data File.

**Site-directed mutagenesis**. Catalytic mutant of murine Arginase II plasmid (pCMV-SPORT6-Arg-2) were obtained by a two-step PCR using the QuickChange site-directed mutagenesis kit (Agilent, 210158) using primers indicated in Supplementary Table 2. Following mutagenesis, the product was combined with competent *E.coli* XL10 Gold cells (100 μL) on ice and incubated for half hour. Following heat shock at 42 °C for 30 s, cells were re-incubated on ice for 2 min, with subsequent addition of pre-warmed LB broth and incubation at 37 °C, shaking for 1 hour to allow recovery prior to streaking on LB-Agar plates and growth overnight. Following overnight growth, colonies were selected and streaked on a

grid plate as well as used as template DNA for a colony PCR. After emergence of colonies, a single colony was selected for purification, followed by growth in a starter culture and inoculation into 150 ml and midi prep of the plasmid using Plasmid Midi Kit (Qiagen, 12143) as per the manufacturer's instructions. Mutant was confirmed using Allele-specific oligonucleotide (ASO) PCR and sequencing from Source Bioscience. For ASO-PCR, a KAPA2G master mix was made for the wild-type and mutant in PCR tubes. 1 ng/μl plasmid DNA was added to the master mix. The PCR product was diluted in Agarose gel (6×) glycerol based sample buffer and ran on a 2% (w/v) agarose gel to confirm mutagenesis.

**Confocal microscopy**. BMDMs were plated at $5 \times 10^5$ seeding density on μ-Dish 35 mm, high Glass Bottom (Ibidi, Germany) and maintained overnight at 37 °C in a 5% CO₂ atmosphere. Cells were stimulated at concentrations as described earlier. After 24 h stimulation, cell samples (from individual wells) were taken for imaging. Thirty minutes before the indicated time, cells were loaded with MitoTracker Red CMXRos (Life technologies) at a final concentration of 50 nM. Cells were washed with DPBS, replaced with fresh complete DMEM and observed in a Leica SP8 scanning Confocal (Wetzlar, Germany), with a ×63 immersion objective. Images were analysed with the Image-J software (National Institutes of Health, Bethesda, MD). For Image-J analysis, mitochondrial length of more than 50 mitochondrial particles per cell was measured in over 25 cells per experimental condition, out of three independent experiments. Mitochondria were then divided into three different categories, based on length, as mitochondria of less than 1 μm, 1–3 μm, and greater than 3 μm, as described by ref. [30].

**Luciferase assay**. The full-length mouse Arg2 3′UTR (270 bp) was amplified using Q5 High-Fidelity DNA Polymerase (NEB) and inserted into XhoI-digested pmir-GLO vector (Promega) using the GenBuilder Cloning Kit (Genscript). Plasmids were isolated from bacterial cultures with the Plasmid Midi Kit (Qiagen, 12143) and the fidelity of the resulting construct (i.e., pmir_Arg2) was confirmed by sequencing. The sequences of cloning and sequencing primers are reported in Supplementary Table 2.

Raw 264.7 cells were seeded in a 96-well plate at a final density of 80,000 cells/well and incubated for 24 h. Cells were then co-transfected with 100 ng of pmir_Arg2 and either mirVana™ miR-155-5p mimic (25 nM, ID MC13058, ThermoFisher Scientific) or inhibitor (50 nM, ID MH13058, ThermoFisher Scientific) in serum-free DMEM using TransIT-X2 Dynamic Delivery System (Mirus) as transfection reagent. Luciferase activity was assessed at 24 hours after transfection using Dual-Luciferase Reporter Assay (Promega, E1910) according to the manufacturer's instructions. RLU (relative luciferase units) expressed as mean value of the firefly luciferase/Renilla luciferase ratio of three independent experiments performed in triplicate were used for statistical analyses.

**Real-time polymerase chain reaction**. Total RNA was isolated using the Qiagen RNeasy Plus Mini kit (Sigma, 74106) and quantified using a Nano-Drop 1000 Spectrophotometer (Thermo Scientific Fisher). cDNA was prepared using 50–100 ng/μl total RNA using a high capacity cDNA reverse transcription kit (Applied Biosystems, 4368814), according to the manufacturer's instructions. Primers were designed using the NCBI database (https://ncbi.nlm.nih.gov/tools/primer-blast) and provided by Sigma. Please refer to Supplementary Table 2 for list of primers used. Real-time quantitative PCR (RT-PCR) was performed on cDNA, diluted 1 in 2 with RNAase-free water, using SYBR Green probes on a 7900 HT Real-Time PCR System (Applied Biosystems). Fold changes in expression were calculated by the Delta-Delta (ΔΔ) Ct method using *18s* as a control for mRNA expression. All fold changes were normalised to untreated/non-targeting controls.

**Cytokine measurement**. For cytokine measurements, cells were stimulated as indicated and supernatants removed and analysed for mouse IL-6, TNF, IL-10 and IL-1β using Enzyme-linked Immunosorbent Assay (ELISA) ((DuoSet, R&D, DY406 (IL-6), DY410 (TNF), DY417-05 (IL-10), DY401-05 (IL-1β)) with absorbance read using a microplate reader at 450 nm and compared against a standard curve. Alternatively, the cytometric bead array (CBA), mouse inflammation kit (BD Biosciences, 339199) was used to examine a panel of cytokines for in vivo experiments, according to manufacturers' instructions. Sample data for CBA was acquired using Attune NxT Flow Cytometer with recommended voltage settings for PE and APC-Cy7.

**Arginase assay**. Arginase activity was determined in BMDM and Raw 264.7 using the Arginase Activity Assay Kit (Sigma, MAK112-1KT) following manufacturer's protocols. Briefly, cells were lysed in low-stringency lysis buffer (provided in kit) followed by the addition of arginine substrate buffer (provided in kit) and incubated at 37 °C for 2 h, after which time the reaction was stopped by the addition of Urea Reagent. Plate was incubated for a further 1 h at room temperature, followed by determination of absorbance at 430 nm. Arginase activity in μmol/minute was calculated as directed by the manufacturer based on the absorbance of a 1 mM urea standard. Specific activity (μmol/min/mg) was calculated following the determination of protein concentration in cell lysates by BCA assay (ThermoFisher Scientific, 10678484).

**Greiss assay**. NO production was measured using the Griess Reagent (Sigma) by addition of reagent to sample supernatants in a 1:1 ratio. Absorbance in culture media was detected by a plate reader at 540 nm and compared against a standard curve.

**Fumarate assay**. BMDM were plated in 6-well plates at $3 \times 10^6$ cells/well in complete DMEM and 10% (v/v) L929 media. Cells were collected by centrifugation and sonicated for 10 s at amplitude 35% in an appropriate volume of cold buffer containing 50 mM potassium phosphate (pH 7.5) and protease inhibitor cocktail tablet (Roche). The assay was then performed according to the manufacturer's instructions of the ChromaDazzle Fumarate Assay kit (AssayGenie, BA0110) and absorbance was read at 565 nm.

**Succinate assay**. BMDM were plated in 6-well plates at $3 \times 10^6$ cells/well in complete DMEM and 10% (v/v) L929 media. Cells were collected by centrifugation and lysed in an appropriate volume of assay buffer provided and protease inhibitor cocktail tablet (Roche). The cells were centrifuged at $10,000 \times g$ for 5 min and supernatant was collected for assay. The assay was then performed according to the manufacturer's instructions of the Assay kit (Abcam, ab204718) and absorbance was read at 450 nm.

**Flow cytometry**. BMDMs were seeded in non-tissue-culture plates and stimulated as described above. Cells were stained with MitoTracker Green (for total mitochondrial mass) and MitoTracker Red CMXRos (for mitochondrial membrane potential) or MitoSOX (for mitochondrial ROS) according to manufacturer's instructions (Invitrogen) in combination with Live/Dead stain (eBioscience™ Fixable Viability Dye eFluor™ 780). For flow cytometry analysis, data (10,000 events per sample) was acquired with the Attune NxT Flow Cytometer and analysed with FlowJo v10 software (TreeStar). A flow gating strategy is provided in Fig. 5 of Supplementary Information.

**ImageStream**. $1–2 \times 10^6$ BMDM or Raw 264.7 cells were collected in a round-bottom 96-well plate, washed with ice-cold DPBS and resuspended in 100 µl of 4% paraformaldehyde (PFA) fixation buffer. Cells were incubated at 37 °C for 15 min, after which they were washed twice with PBS-T (10 min incubation between washes). This was followed by resuspension in ice-cold methanol and incubation at −20 °C for 15 min. The cells were washed again twice with PBS-T, and resuspended in blocking buffer (5% goat serum; 0.3% Triton-X100 in PBS) at room temperature for 1 h. The cells were briefly spun, the blocking buffer was removed and the antibody cocktail was added as required (1:100 or 1:200 dilution in blocking buffer), and incubated overnight at 4 °C. The following day the cells were spun, rewashed three times with PBS-T, and Alexa-fluor fluorescent secondary antibody cocktail was added (diluted in blocking buffer at 1:200 concentration). The cells were incubated with the secondary antibody for 1 h in the dark at RTP, after which they were washed twice with PBS-T. Finally, the cells were resuspended in 50 µl of PBS with 600 nM DAPI (Invitrogen). For imaging, an ImageStream Mk-II cytometer (Amnis Inc, Seattle, USA) was used. The ImageStream was equipped with 488, 658, and 405 nm laser sources with variable laser power (from 20 to 200 mW for the 488 nm laser and from 20 to 100 mW for the 658 nm laser) and a brightfield light source. Files of 1000–5000 events were acquired for each sample and 100–200 events were acquired for single fluorochrome controls. Additional single colour control files were collected in the absence of brightfield illumination for use in creating the compensation matrix with IDEAS software (Amnis, Seattle, USA). The calculated compensation matrix was applied to all files to correct for spectral crosstalk. The resulting compensated cytometry data were further analysed with the IDEAS software program. Gating of cell events with the area and aspect ratio was used to eliminate debris (low area) and multi-cellular events (large area, high aspect ratio) from further analysis. After defining single, focused cells, a colocalization program was run using the IDEAS software to check for localization of Arg2 at the mitochondria by gating an Arg2 and Tom20 double-positive population. The software was then set to plot results for median bright detail similarity (BDS) and compared to the BDS of a control sample containing colocalization score for two known mitochondrial proteins (Tom20 and Hsp60).

**Metabolic flux analysis**. Cells were plated at $5 \times 10^4$ cells/well (BMDMs) and $3 \times 10^4$ cells/well (Raw 264.7) on an XF$_e$96-well plate (Agilent) in complete DMEM (+10% L929 supernatant in media for BMDM). Cells were treated and stimulated accordingly. A utility plate containing the injector ports and probes was filled with calibrant solution (200 µl/well) and placed in a CO$_2$-free incubator at 37 °C overnight. Following 24 h stimulation, complete DMEM was removed from cells and replaced with XF assay media pH 7.4 (Agilent) at 175 µl/well. The XF assay media was supplemented with 10 mM glucose, 1 mM pyruvate and 2 mM glutamine (Sigma). The cell culture plate was then placed in a CO$_2$-free incubator for 45 min. Oligomycin (ATPase inhibitor, 1 µM), Carbonyl cyanide-p-trifluoromethoxyphenylhydrazone (FCCP) (0.9 µM) and Rotenone/Antimycin A (0.5 µM) were added to the appropriate ports of the utility plate for a standard MitoStress test according to instruction manual (Agilent, 103015-100). This plate was run first on the flux analyser for calibration. Once complete, the utility plate

was replaced with the cell culture plate and run on the real-time Seahorse XF$_e$96 analyzer for 1 h 30 min using the software's MitoStress template program. The OCR values for Raw 264.7 were normalised using absorbance readings obtained from a protein quantification BCA assay (Pierce). Data were analysed using the Wave software.

**Complex-mediated respiration**. For customised assays to measure individual complex I (NADH-linked) and complex II (FADH$_2$-linked) specific respiration status in intact mitochondria, cells were resuspended in MAS buffer (70 mM sucrose, 220 mM mannitol, 10 mM KH$_2$PO$_4$, 5 mM MgCl$_2$, 2 mM HEPES and 1 mM EGTA, and 0.2% fatty-acid free BSA). The utility plate injector ports were loaded with complex I or complex II specific substrates. Final concentrations/well of complex-I specific substrates was 2.5 mM glutamate and 2.5 mM malate; and complex-II specific substrate was 2.5 mM succinate (all from Sigma). After calibration of utility/injector plate, right before inserting cell-plate in the machine, cells were supplemented with 4 mM ADP, 0.5 µM Rotenone and 10 µg/ml digitonin. The assay was run for 30 min and included three initial pre-injection measurements and three post-injection measurements. Changes in OCR upon substrate injection were calculated as an increase in percentage from pre-injection OCR levels that were at baselined at 100%.

**MTT assay**. BMDMs were seeded at $5 \times 10^5$ cells/ml in 96-well plates and left unstimulated or stimulated with 100 ng/mL LPS or LPS + IL-10 for 24 h. Succinate dehydrogenase activity was assessed by yellow 3-(4,5-dimethylthiazol-2-yl)-2,5-diphenyltetrazolium bromide (MTT) reduction to purple formazan. Cells were incubated with 1 mg/mL MTT (Sigma) for 3 h at 37° C and 5% CO$_2$ and subsequently, the media was aspirated off so the cells could be lysed with dimethyl sulfoxide (DMSO). Absorbance of soluble formazan was measured spectrophotometrically at 590 nm and background signal measured at 650 nm was subtracted from this value for each sample. Cells were also treated for 24 hours with 10 mM of the SDH inhibitor dimethyl malonate (DMM) (Sigma) were included as a control.

**Spectrophotometric CII specific activity**. BMDM stimulated as indicated and cells were scraped and centrifuged at $200 \times g$ for 5 min. Cell pellets were homogenized using a glass 2 ml dounce homogenizer with 25 strokes before freeze thawing of lysates three times using liquid nitrogen. For CII activity, between 0.1 and 0.5 mg of cell lysate was added to a 1 ml cuvette containing 25 mM phosphate, 1 mg/ml fatty acid–free BSA, 0.3 mM KCN, 20 mM succinate and 75 µM 2,6-Dichlorophenolindophenol (DCPIP) (Sigma). Samples were incubated for 10 min at 37 °C before reading baseline absorbance at 600 nm using a UV-VIS spectrophotometer. Baseline measurements were taken for 3 min before the addition of 50 µM decylubiquinone (Sigma). The ΔOD was followed for 4–5 min before the addition of 20 nM Atpenin A5 (Sigma), a specific inhibitor of CII, to observe the CII insensitive rate. CII specific activity was calculated according to the Beer–Lambert law using a molar extinction coefficient for DCPIP of 19.1 mM cm$^{-1}$. Citrate synthase activity was measured to account for mitochondrial content of samples and to normalize CII enzymatic rates. For citrate synthase activities, 0.1–0.5 mg cell lysate was added to a 1 ml cuvette containing 100 mM Tris pH 8.0 with 0.2% V/V Triton X-100, 100 µM DTNB (ThermoFisher Scientific), and 300 µM Acetyl CoA (Sigma). Baseline absorbance was read at 412 nm for 3 min before initiating the reaction by the addition of 500 µM oxaloacetate. The ΔOD was followed for 4 min. Citrate synthase specific activities were calculated according to the Beer–Lambert law using the molar extinction co-efficient of DTNB of 13.6 mM cm$^{-1}$. CII specific activity was expressed as a ratio to citrate synthase specific activity.

**Statistical analysis**. GraphPad Prism 8.00 (GraphPad Software) was used for statistical analysis. A one-way ANOVA test was used for the comparison of more than two groups, with Tukey or Sidak's test for multiple comparisons. Analysis of data with two or more factors were analysed by a two-way ANOVA with the Sidak's test for multiple comparison. A two-tailed Student's $t$-test was used when there were only two groups for analysis. By visual checking of normality distribution plots, either parametric tests were performed, or Mann–Whitney nonparametric equivalents were employed where Shapiro–Wilks distribution was violated ($p > 0.05$). All error bars represent SEM. Significance was defined as *$p < 0.05$, **$p < 0.01$, ***$p < 0.001$, ****$p < 0.0001$. Any specific statistical tests and details of 'n' numbers done for experiments are listed under the corresponding figures.

**Reporting summary**. Further information on research design is available in the Nature Research Reporting Summary linked to this article.

## Data availability

The gene expression datasets generated by the Affymetrix array have been deposited in GSE under accession code GSE151835. The authors declare that all other data are available in the article and Supplementary information files. Source data are provided with this paper.

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

## Acknowledgements

The majority of this study was supported through the funding provided by Science Foundation Ireland (SFI16/FRL/3855) for C.E.M., J.K.D. and R.A., as well as contributions from National Health and Medical Research Council Australia (APP1063400); for M.P.C.S. and A.C. from CONACYT (CVU440823) and Science Foundation Ireland Career Development Award (17/CDA/4688); for N.A. and D.K.F. from ERC-CoG (770769 DC_Nutrient); for G.R.B. and C.H. from Swiss National Science Foundation (SNSF31003A_172848); for D.J.G. from the Children's Cancer Foundation and a Victoria Cancer Agency Mid-Career Fellowship (MCRF19033); for C.D.S. and C.D. from the Irish Research Council (GOIPD/2018/575 and GOIPG/2018/2648); for A.L.G., A.M.L., E.C.L. from Science Foundation Ireland (SFI12/IA/1421), Irish Research Council (GOIPG/2015/4023) and EPSPG/2017/302; and for S.A. and T.R. National Children's Research Centre (C/18/9).

## Author contributions

J.K.D. and R.A. designed and performed the majority of experiments, analysed the data, wrote and edited the manuscript. L.J.G., P.J.H. and B.R.W. conducted bioinformatic analysis. S.A. and T.R. performed in vivo LPS experiments with *Arg2*$^{-/-}$ animals. G.M.D. and G.P.D. performed and analysed the complex-II spectrophotometric assay experiments. M.P.C.S. and A.M.C. generated and analysed confocal microscopy data. C.D.S. performed luciferase reporter experiments and assisted in conducting experiments with THP-1 cells and in vivo LPS experiments. N.A., K.D., D.K.F. and P.J.O. generated and analysed the GC/MS data. D.J.G. performed the mitochondrial import experiments. G.R. B. and C.H. aided with the ImageStream acquisition and analysis. F.N., F.I.H. and C.P.D. assisted in conducting experiments. A.L.G., A.M.L. and E.C.L. provided *Il10*$^{-/-}$ mice and performed in vivo LPS experiments. C.E.M. led the project, acquired the funding for project, designed experiments, wrote and edited the manuscript.

## Competing interests

The authors declare no competing interests.
