## [Peer Review File · Nature Communications]

REVIEWER COMMENTS

Reviewer #1 (Remarks to the Author):

In this manuscript, Dowling and colleagues present evidence that Arg2 has key functions in regulating the inflammatory and anti-inflammatory responses to LPS and IL-10, respectively. The authors demonstrate that Arg2 activity increases in response to IL-10 and being a mitochondrial isoform of arginase, regulates mitochondrial fission and fusion (Fig. 2f). Important experiments concerning the bioenergetic effects of Arg2 are shown in Fig. 3. Overall, the manuscript helps explain previous work from the Medzhitov group on IL-10-mediated regulation of TLR-induced metabolic changes, and provides new and exciting information about the role of Arg2.

The major comment concerns the appearance of some of the gels which seem to have technical issues (1h, 2a, 2b for tubulin, 4j, especially) and the bands are not sharp. The related issue concerns the resolution of the mitochondrial morphology issues in 2d-f where the images are not particularly clear.

Reviewer #2 (Remarks to the Author):

It is well appreciated that programming of mitochondrial bioenergetics and dynamics is an essential factor in the regulation of immune cell function. LPS treatment is known to drive mitochondrial fragmentation and glycolysis in activated macrophages, while IL-10 can oppose this metabolic shift by promoting OXPHOS (Eddie IP et al., Science 2017). This interesting study by Dowling et al. has explored how mitochondrial metabolism regulates macrophage polarisation in response to the anti-inflammatory cytokine IL-10. They confirm that IL-10 promotes mitochondrial respiration, which they associate with an elongated mitochondrial network. Mechanistically, they convincingly show Arginase-2 (Arg2) to be upregulated by IL-10 and demonstrate that Arg2 is required for IL-10-mediated mitochondrial elongation and OXPHOS. Less convincingly, they propose that mitochondrial Arg2 somehow influences SDH activity, a key regulator of macrophage metabolic reprogramming (e.g. Mills et al., Cell 2016). They further show that Arg2 is required for the IL-10 mediated suppression of ROS and IL-1 β . While Arginase activity downstream of IL-10 has been previously explored, this is a novel study with several interesting observations that puts forward a putative role for mitochondrial Arg2 in macrophage metabolism and polarisation.

Comments:

- The authors show that Arg2 catalytic activity is required for complex II-dependent respiration (Fig. 4). How this may occur is not addressed in the text unfortunately and it may be too early to state that Arg2 regulates SDH activity. While this may be for future work, the lack of discussion leads to a confusing end to the manuscript. Are the authors proposing that this is the primary effect of Arg2 overexpression, which ultimately drives / facilitates mitochondrial elongation? To unequivocally show that mitochondrial Arg2 function is required for upregulated OXPHOS and SDH function in macrophages, the authors could include the overexpression of Arg2 lacking its mitochondrial targeting sequence in experiments such as Fig 3h.
- The Arg2 mitochondrial localization data in Fig 2 must be improved in order to convince. The import assay should be performed as a time course and include a depolarized mitochondrial control. It is unclear what the red asterisk is referring to and the bands require better labelling. The authors write that Arg2 is "at mitochondria" but they could be more specific. Do they propose that it is imported into the matrix? The crude mitochondrial fractionation in Fig 3b requires mitochondrial and cytoplasmic fractions to be ran on the same membrane and should include other organelle markers e.g. ER. Finally, the resolution of the ImageStream data prevents Fig. 3c from convincingly showing mitochondrial localization. Can the authors use confocal microscopy as they have nicely done for morphology analysis in Fig. 2 d etc?

- Can the authors explain why mitochondrial pellet Succinate levels are increased after LPS+IL-10 treatment (Fig S4a), as well as Fumarate and Malate (Fig 4a)? Is it possible to also determine the Arg2-relevant metabolites in mitochondria i.e. Arginine/ornithine to show increased activity of mitochondrial Arg2 upon LPS+IL-10 treatment? This is intriguing because depletion of Arg1 and Arg2 cause a similar inhibition of whole cell urea production (Fig s1j), leading us to question further how the effect on mitochondrial function can be specific to Arg2.

Other comments:

- Some of the data presentation could be improved to help the reader interpret the data. In Fig. 1 b, can the authors clearly define miR-155 and Arg2 and label some of the other common upregulated and downregulated genes labelled in blue and red? What does the green circle refer to? Where is Arg1 here?
- The Mitotracker red CMXRos figure in Fig.2 h should include the entire data set shown in Fig S2h. Though statistically different, the effect is very minor and this is missed in Fig 2 h due to the y-axis beginning at 50000.
- I think it is unhelpful to split some of the data between main figure and supplement. The OCR traces from control BMDMs should be included in Fig. 3 for example.
- Fig 3 should also include better labelling to show which genotype is being measured in each experiment.
- Please check referencing. e.g. Line 214-215 "In support of this finding, a positive correlation between systemic IL-10 levels in 175 healthy human patients and genes involved in mitochondrial fusion was previously reported".

Reviewer #3 (Remarks to the Author):

The manuscript by Dowling and colleagues propose Arginase-2 as a downstream mediator of IL-10 in activated macrophages. They show that Arg2 is crucial in skewing the mitochondrial bioenergetics into an oxidative state through inducing the activity of ETC complex II (or succinate dehydrogenase – SDH). This is a well-written manuscript with interesting insights in terms of the mitochondria-driven polarization of macrophages in mice. The manuscript provides further mechanistic insights into the recently described anti-inflammatory effect of IL-10 mediated by metabolic reprogramming of macrophages published by Medzhitov and colleagues (Ip et al., reference 18). However the findings lack a disease context and validation in human cells. Some results are a replication of Ip et al. paper, which takes away the novelty factor. Metabolomics and ETC results in Figure 4 are not interpreted correctly and the presented data do not support the conclusions due to lacking experimental conditions. Specifically:

1. Exogenous addition of IL-10 to LPS-stimulated macrophages M(LPS+IL10): LPS will cause an increase in endogenous IL-10 and the further addition of IL-10 seems artificial, without mimicking any in vivo state of activation of macrophages. I understand that the usage of IL10^{-/-} mice controls for the endogenous IL-10 and the authors have a previous paper on miR-155/IL-10 pathway. However, the specific conditions that trigger Arg2 in BMDMs (LPS +IL-10) are context and (possibly) specie-specific. The manuscript will benefit from acknowledging and discussing this throughout. This is important as most of the significant results on the mitochondrial effect of Arg2 (e.g. mitochondrial fusion, membrane potential) are obtained using this specific condition.
2. Were the Arg1^{+/-} mice viable and showed decreased Arg1 and urea in M(LPS+IL-10)?
3. Figure 2b convincingly shows the mitochondrial localization of Arg2 in BMDMs stimulated with LPS+IL-10. Is the same true in human macrophages?
4. What is the effect of mir-155 on the mitochondrial effects of Arg2?
5. Figure 3a-b is a replication of previously published results by Ip et al. (reference 18). The siRNA experiment was done in Raw cells and should be replicated in BMDMs for accurate comparison with

the Arg2 KO mice BMDMs.

6. Figure 4a: If fumarate and malate levels are the same between LPS and LPS+IL-10 conditions, then it would not be correct to claim that LPS+IL-10 is causing a specific metabolic shift. Statistics should be done between LPS and LPS+10 and in case it is not significant, the results should be re-considered.

7. Figure 4b: The LPS condition should be added – the comparison between LPS+IL-10 and CTRL is only partially informative. Same with Figure 4e.

8. Figure 4g. The over-expression of Arg1 should also be investigated in terms of CII activity.

9. The in vivo data (Figure 4k-m) is premature and show only mRNA levels of cytokines. Figure 4k shows IL-1beta mRNA levels (right panel) whereas Figure 4K also shows mRNA levels of IL-1beta but this time significant. This part of the paper is significantly weak and needs to be reconsidered in view of an appropriate in vivo model such as colitis model in Il10^{-/-}-mice.

Rebuttal

We thank the reviewers for their thorough assessment of our manuscript. As detailed below we have addressed reviewers comments by performing additional experiments where requested and also clarified and improved the text. Thank you again for taking the time to review this work, as the manuscript is significantly enhanced by the additional work that was suggested.

Reviewer #1 (Remarks to the Author):

In this manuscript, Dowling and colleagues present evidence that Arg2 has key functions in regulating the inflammatory and anti-inflammatory responses to LPS and IL-10, respectively. The authors demonstrate that Arg2 activity increases in response to IL-10 and being a mitochondrial isoform of arginase, regulates mitochondrial fission and fusion (Fig. 2f). Important experiments concerning the bioenergetic effects of Arg2 are shown in Fig. 3. Overall, the manuscript helps explain previous work from the Medzhitov group on IL-10-mediated regulation of TLR-induced metabolic changes, and provides new and exciting information about the role of Arg2.

We thank the reviewer for their positive comments and for stating that this work provides new and exciting information about the role of Arg2.

The major comment concerns the appearance of some of the gels which seem to have technical issues (1h, 2a, 2b for tubulin, 4j, especially) and the bands are not sharp.

We would like to thank the reviewer for noticing this issue on our behalf. We have amended the figures to include westerns from other replicates for the experiments concerned, where the tubulin is better and the bands are sharper, please see as follows:

- For **Fig. 1h**, westerns have been updated and subsequent to revision have been moved to **Fig. S1g**
- **Fig. 2a** –This is an import assay for which tubulin is not a requirement. Essentially, after the import assay is done we run an SDS-PAGE and then dry the gel for imaging with a phosphoimager. We do not transfer to membrane. However, we do quantify the mitochondrial pellet and use the same mass in each import assay (typically 50-100ug)
- For **Fig. 2b** the western was sharpened to enhance resolution for tubulin
- For **Fig. 4j** the westerns have been updated with a new replicate and this is now found in **Fig. S4e**

The related issue concerns the resolution of the mitochondrial morphology issues in 2d-f where the images are not particularly clear.

*We have gone back to our images looking at mitochondrial morphology in **Fig. 2d-f** and made them as clear as possible by enlarging them in the figure and sharpening resolution.*

Reviewer #2 (Remarks to the Author):

It is well appreciated that programming of mitochondrial bioenergetics and dynamics is an essential factor in the regulation of immune cell function. LPS treatment is known to drive mitochondrial fragmentation and glycolysis in activated macrophages, while IL-10 can oppose this metabolic shift by promoting OXPHOS (Eddie IP et al., Science 2017). This interesting study by Dowling et al. has explored how mitochondrial metabolism regulates macrophage polarisation in response to the anti-inflammatory cytokine IL-10. They confirm that IL-10 promotes mitochondrial respiration, which they associate with an elongated mitochondrial network. Mechanistically, they convincingly show Arginase-2 (Arg2) to be upregulated by IL-10 and demonstrate that Arg2 is required for IL-10-mediated mitochondrial elongation and OXPHOS. Less convincingly, they propose that mitochondrial Arg2 somehow influences SDH activity, a key regulator of macrophage metabolic reprogramming (e.g. Mills et al., Cell 2016). They further show that Arg2 is required for the IL-10 mediated suppression of ROS and IL-1 β . While Arginase activity downstream of IL-10 has been previously explored, this is a novel study with several interesting observations that puts forward a putative role for mitochondrial Arg2 in macrophage metabolism and polarisation. The authors show that Arg2 catalytic activity is required for complex II-dependent respiration (Fig. 4). How this may occur is not addressed in the text unfortunately and it may be too early to state that Arg2 regulates SDH activity.

We thank the reviewer for their positive comments and are glad they have found the study and findings surrounding the role of Arg2 in macrophage interesting and novel. We agree fully that the observation with regard to SDH activity warranted further investigation and full validation. We have completed several additional experiments to strengthen and confirm this finding.

We have now included new data which measured the reduction of 3-(4,5-dimethylthiazol-2-yl)-2,5-diphenyltetrazolium bromide (MTT) to formazan, a common assay for SDH activity as reported by others (Koenis, D. S. et al., Cell Reports 2018; Van den Bossche et al., Cell Reports 2016). This assay demonstrated that Arg2^{-/-} BMDMs had lower levels of SDH activity in both resting and (LPS+IL-10) cells compared to Arg2^{+/+} BMDMs (Fig 4f, lines 208-211, page 5). We also used the MTT assay in Raw264.7 macrophages transfected with Arg2 overexpression plasmid and compared it to the catalytic mutant H145F and an empty vector control to validate that boosting Arg2 also boosts SDH activity (Fig S4c, lines 214-215, page 5).

In a second set of experiments, we used a sophisticated assay to specifically measure CII activity using a single wavelength spectrometer technique. CII specific activity is measured as a ratio to citrate synthase (CS) specific activity, in order to normalize activities to mitochondrial mass as previously described (Spinazzi, M. et al., Nat Protocols 2014). We demonstrate that LPS reduces CII activity, which is restored in the presence of IL-10 in Arg2^{+/+} (Fig 4g, lines 215-221, page 5-6). This ability of IL-10 to restore CII activity was lost in Arg2^{-/-} BMDM. Results also demonstrated that CII specific activity was significantly decreased in resting Arg2^{-/-} cells compared to Arg2^{+/+} controls.

Finally, we used a commercialized succinate measuring assay kit (Abcam) to spectrophotometrically measure the amount of succinate in our wild-types vs Arg2 deficient BMDM, and found that whilst addition of IL-10 was able to bring down accumulated succinate levels in WT inflammatory macrophages, this was not the case in an Arg2 deficient BMDM LPS+IL-10 treatment. (Fig. 4h, lines 223-225, page 6)

Together, we hope this additional data convinces that Arg2 upregulates SDH activity.

References:

Koenis, D. S. et al., Nuclear Receptor Nur77 Limits the Macrophage Inflammatory Response through Transcriptional Reprogramming of Mitochondrial Metabolism Cell Reports (2018)
<https://www.ncbi.nlm.nih.gov/pubmed/30134173>

Van den Bossche, J., Mitochondrial Dysfunction Prevents Repolarization of Inflammatory Macrophages. Cell Reports (2016)
<https://www.ncbi.nlm.nih.gov/pubmed/27732846>

While this may be for future work, the lack of discussion leads to a confusing end to the manuscript. Are the authors proposing that this is the primary effect of Arg2 overexpression, which ultimately drives / facilitates mitochondrial elongation?

We have clarified the end of the manuscript by providing more detail and discussion around our key findings.

1. We discuss the role of Arg2 and its impact on mitochondrial dynamics and OxPhos, highlighting that its catalytic activity (i.e. production of ornithine) may be central to this process.

Discussion, Page 6 -7, Lines 258-268

“We show that IL-10 enhancement of Arg2 favors mitochondria in a state of fusion. In support of this finding, a positive correlation between systemic IL-10 levels in 175 healthy human patients and genes involved in mitochondrial fusion was previously reported³⁰. Certainly, mitochondrial dynamics, specifically fusion, has been shown to influence OxPhos^{31,32}. This is linked with our other main finding that Arg2 facilitates oxidative respiration via its impact on CII activity at the ETC. Interestingly we found these effects to be, at least in part, dependent on Arg2’s catalytic activity, suggesting ornithine production may play a key role in this process. Previous work by others has highlighted that IL-4 upregulation of ornithine is essential for hypusination of the eukaryotic translation initiation factor 5a (elf5a), which works to maintain integrity of mitochondrial TCA cycle and OxPhos proteins³³. Whether Arg2’s impact on mitochondrial respiration is dependent on the production of ornithine requires further investigation.”

2. We discuss implications for Arg2 increasing SDH activity and restraining succinate levels. In doing so, IL-10 boosts fumarate production via Arg2, and is able to regulate Hif-1a stabilization and IL-1β secretion.

Discussion, Page –7, Lines 270-285.

“Specifically, LPS induced inflammatory macrophages increase their succinate levels with decreased alpha-ketoglutarate levels. Work from Baseler et al., demonstrated that the LPS-induced accumulation in succinate in wild-type BMDM was abolished in IL-10^{-/-} BMDM, yet the reductions of alpha-ketoglutarate, indicative of the LPS-induced “TCA break”, remained completely intact²². This suggests that autocrine IL-10 acts to relieve the TCA cycle “break” by increasing metabolites downstream of succinate, such as fumarate and malate. In that regard, SDH, a central enzyme of the TCA cycle, converts succinate to fumarate, and is shown to be inhibited in LPS-stimulated macrophages^{27,35}. Adding to these studies, we

have shown that IL-10 decreases succinate, enhances SDH/CII activity and increases fumarate production in inflammatory macrophages, an effect we demonstrate to be dependent on Arg2. Moreover, we show that in Arg2^{-/-}, HIF-1 α and IL-1 β expression is elevated. This is interesting considering excess succinate can lead to HIF-1 α stabilization²⁸, while patients harboring mutations in SDH have increased HIF-1 α activity^{36,37} and circulating succinate levels³⁸. Collectively, our work demonstrates that Arg2 is integral in resolving the inflammatory status of the cell.”

3. We discuss how fumarate has been shown to have therapeutic potential in the form of Dimethyl fumarate (DMF). Interestingly, we highlight how the mechanistic efficacy of DMF mirrors the effects we see with enhanced Arg2 in macrophages.

Discussion, Page –7, Lines 286-293.

“Curiously, the fumarate analogue Dimethyl Fumarate, DMF, is an immunomodulatory drug used for the treatment of inflammatory disorders like psoriasis and multiple sclerosis (MS). Furthermore, monocytes from DMF-treated relapsing-remitting MS patients have decreased expression of the pro-inflammatory micro-RNA miR-155. DMF has also been shown to reduce pro-inflammatory cytokines such as IL-1 β enhance mitochondrial oxidative respiration, and boost the anti-oxidant Nrf2 pathway. In our studies, we show a similar profile with Arg2, where we observed enhanced Arg2 in miR-155^{-/-} macrophages. Furthermore, a loss of Arg2 in LPS-stimulated macrophages results in increased mtROS, Hif-1 α and IL-1 β secretion, along with reduced Nrf2 expression, despite the presence of IL-10.”

To unequivocally show that mitochondrial Arg2 function is required for upregulated OXHPOS and SDH function in macrophages, the authors could include the overexpression of Arg2 lacking its mitochondrial targeting sequence in experiments such as Fig 3h.

Unfortunately, due to COVID19 restrictions we could not generate an MTS mutant for ARG2 in time, which we agree would have been beneficial to this study. Instead, we have included Arg1 overexpression (the arginase isoform which does **not** have an MTS, and is cytosolic) and hence, acts as relevant negative control. In addition, we provide new data to confirm Arg2’s role in regulating OxPhos via SDH activity as follows:

1. Arg1 overexpression cannot boost Oxphos and CII specific OCR (**Fig 3g-h and 4e**).
2. Arg2 H145 mutant cannot boost CII OCR (**Fig 4e**)
3. We used 3 different assays to confirm Arg2 regulation of SDH by performing an MTT assay in WT and Arg2 deficient BMDM (**Fig 4f**), as well as in Raw264.7 macrophages overexpressing Arg2 and the H145F catalytic mutant (**Fig. S4c**). Moreover, we used a sophisticated spectrophotometric assay looking at Complex II specific activity normalised to mitochondrial mass (**Fig 4g, lines 215-221**), in addition to our OCR-based assays. Finally, we utilised a succinate assay kit to measure succinate levels between WT and Arg2 deficient BMDM to show Arg2 knock-out cells cannot reduce their succinate levels despite presence of IL-10 in an inflammatory setting (**Fig. 4h, lines 223-225, page 6**)

The Arg2 mitochondrial localization data in Fig 2 must be improved in order to convince. The import assay should be performed as a time course and include a depolarized mitochondrial control. It is unclear what the red asterisk is referring to and the bands require better labelling. The authors write that Arg2 is “at mitochondria” but they could be more specific. Do they propose that it is imported into the matrix?

To improve the localisation experiments surrounding Arg2 we have performed a time course experiment and included a depolarized mitochondrial control (+FCCP) as suggested by the reviewer. The data shows time points of 15 min, 30 min, 1 h and a 1h +FCCP as a depolarisation control (Fig. 3a, lines 105-113, page 3). It shows very clearly that Arg2 is imported as early as 15 min, and that Arg1 is not imported. Import of Arg2 is not impacted by the loss of mitochondrial membrane potential (i.e. the addition of +FCCP) suggesting that it does not enter the matrix. This was consistent across three independent experiments.

In the original manuscript the red asterisks referred to proteinase K treatment where reactions were treated with proteinase K to digest non-mitochondrial imported proteins. That figure in the original manuscript has now been replaced with the new time course blot as described above.

The crude mitochondrial fractionation in Fig 3b requires mitochondrial and cytoplasmic fractions to be ran on the same membrane and should include other organelle markers e.g. ER.

We performed these experiments to ascertain whether stimulation with LPS alone or with LPS+IL-10 induced Arg2’s translocation to or from the mitochondria. Such translocation has been reported for endothelial cells in response to Oxidized low-density lipoprotein (OxLDL) (Pandey, D. et al., Circ Res 2014). Moreover, these experiments were performed to compare responses in WT vs IL-10 KO and thus samples were run separately as mitochondrial and cytoplasmic fractions. To satisfy the reviewer’s comments, we have included the full blots here to demonstrate our rationale for the design of the blot layout.

However, due to space restrictions, we would like to request that our original blots remain in situ as they highlight the key point most succinctly: that Arg2 is present in the mitochondrial fraction only, an effect which is enhanced in the presence of LPS+IL10.

Finally, the resolution of the ImageStream data prevents Fig. 3c from convincingly showing mitochondrial colocalization. Can the authors use confocal microscopy as they have nicely done for morphology analysis in Fig. 2 d etc? ???

As the reviewer suggested, we have replaced the imagestream LPS+IL-10 representative image with one that has better resolution, and enlarged the representative image presented in Fig 3c to emphatically show the effect of Arg2/Tom20 colocalisation upon LPS+IL-10 stimulation. The purpose of this experiment was to highlight enhancement of mitochondrial localisation with LPS+IL-10. We believe the ImageStream was the most quantifiable method to do this, as we could determine that in untreated cells Arg2 had 44% colocalization with Tom20 compared to 60% with LPS+IL-10 measured over 1000 events done in two independent experiments. Furthermore, we used another mitochondrial protein, Hsp60, as our positive control in this experiment as shown in Fig S2a.

Can the authors explain why mitochondrial pellet Succinate levels are increased after LPS+IL-10 treatment (Fig S4a), as well as Fumarate and Malate (Fig 4a)?

Work from Baseler et al., demonstrated that the LPS-induced accumulation in succinate in wild-type BMDM was abolished in IL-10^{-/-} BMDM, yet the reductions of alpha-ketoglutarate, indicative of the LPS-induced “TCA break”, remained completely intact²². Through that result, they suggested that “autocrine IL-10 may act to relieve the TCA cycle “break” by restraining succinate levels”, and increasing downstream metabolites such as fumarate and malate. These observations were validated in our GC/MS data where we saw that addition of exogenous IL-10 (i.e. LPS+IL-10) resulted in a further increase in these metabolites compared to LPS alone (Fig. S4a). Of course, it was difficult to tease apart the exact contribution of IL-10 in a WT BMDM (due to effects of autocrine IL-10 being produced in LPS-only treated macrophages), hence the significance between LPS and LPS+IL-10 could not be seen. We circumvented this issue then by comparing significance between unstimulated samples vs LPS, and unstimulated samples vs LPS+IL-10. LPS+IL-10 samples gave us significant increases in fumarate and malate, but not succinate. LPS-alone samples didn't give us that significance.

Interestingly, when we looked at cellular succinate and fumarate levels, we could clearly see that in a WT BMDM, succinate was accumulated with LPS but reduced with LPS+IL-10, whereas fumarate was decreased with LPS, and significantly increased with LPS+IL-10, an effect we showed to be dependent on Arg2 (Fig. 4h-i). This led us to surmise that IL-10 is working to downregulate excess (inflammatory) succinate levels by boosting fumarate levels through Arg2. We currently perceive that this is at least partly due to increased SDH activity mediated via Arg2. Future work will need to expand on these findings by looking at other non-mitochondrial sources of succinate/fumarate in WT and Arg2 deficient BMDM using tracing experiments.

Is it possible to also determine the Arg2-relevant metabolites in mitochondria i.e. Arginine/ornithine to show increased activity of mitochondrial Arg2 upon LPS+IL-10 treatment? This is intriguing because depletion of Arg1 and Arg2 cause a similar inhibition of whole cell urea production (Fig s1j), leading us to question further how the effect on mitochondrial function can be specific to Arg2.

We thank the reviewer for this suggestion. Baseler et al demonstrated with their GC/MS analysis that LPS-stimulated macrophages that were lacking IL-10 had depleted cellular ornithine, urea, proline and putrescine levels compared to WT macrophages stimulated with LPS.

Our own current GC/MS data (WT BMDM that were unstimulated or LPS+IL-10 stimulated) also shows upregulation of Ornithine and downstream polyamine metabolites (putrescine and spermidine) in LPS+IL-10 samples (data not shown in manuscript). Indeed, our future studies plan to more specifically delineate the role of Arg2 by performing labelled metabolomics in LPS+IL-10 stimulated WT vs Arg2^{-/-} BMDM.

Other comments:

Some of the data presentation could be improved to help the reader interpret the data. In Fig. 1 b, can the authors clearly define miR-155 and Arg2 and label some of the other common upregulated and downregulated genes labelled in blue and red? What does the green circle refer to? Where is Arg1 here?

We have made amendments to **Fig. 1b**, miR-155 and Arg2 are more clearly defined and we have labelled several other common upregulated and downregulated genes including: Cxcl10, Il6, Vcam1, Tnf, Il1rn, Il4ra and Arg1. We thank the reviewer for pointing out the unnecessary green circle in the plot legend. This has now been removed. As mentioned the Arg1 gene is now labelled on the plot.

The Mitotracker red CMXRos figure in Fig.2 h should include the entire data set shown in Fig S2h. Though statistically different, the effect is very minor and this is missed in Fig 2 h due to the y-axis beginning at 50000.

We have performed additional experiments over the course of generating the revised manuscript and this experiment and the entire data set are now found in **Fig. S2h**. The y-axis for Mitotracker red CMXRos has also been corrected to begin from zero. We thank you for pointing out this oversight.

I think it is unhelpful to split some of the data between main figure and supplement. The OCR traces from control BMDMs should be included in Fig. 3 for example.

Given the amount of data in the manuscript we agree with the reviewer that splitting the data between the main figures and supplemental was unhelpful for the reader. We have now changed this throughout the manuscript and entire data sets are presented completely in the main or supplemental figures.

Fig 3 should also include better labelling to show which genotype is being measured in each experiment.

We agree with the reviewer and each figure is now clearly labelled with the genotype at the top of the data set being measured in each experiment.

Please check referencing. e.g. Line 214-215 “In support of this finding, a positive correlation between systemic IL-10 levels in 175 healthy human patients and genes involved in mitochondrial fusion was previously reported”.

*Thank you for noting the oversight on our part. We have added the correct reference in **Line 259-260, page 6***

de-Lima-Júnior, J. C. *et al.* Abnormal brown adipose tissue mitochondrial structure and function in IL10 deficiency. *EBioMedicine* **39**, 436-447, doi:<https://doi.org/10.1016/j.ebiom.2018.11.041> (2019).

Reviewer #3 (Remarks to the Author):

The manuscript by Dowling and colleagues propose Arginase-2 as a downstream mediator of IL-10 in activated macrophages. They show that Arg2 is crucial in skewing the mitochondrial bioenergetics into an oxidative state through inducing the activity of ETC complex II (or succinate dehydrogenase – SDH). This is a well-written manuscript with interesting insights in terms of the mitochondria-driven polarization of macrophages in mice. The manuscript provides further mechanistic insights into the recently described anti-inflammatory effect of IL-10 mediated by metabolic reprogramming of macrophages published by Medzhitov and colleagues (Ip *et al.*, reference 18). However the findings lack a disease context and validation in human cells.

*We are thankful to the reviewer for stating that they found our results interesting and insightful. We also agree with reviewer that initially our data did lack both a disease context specific to Arg2, as well as human relevance to our findings. We believe our updated manuscript has benefited greatly from validating our initial murine data in human macrophages, and ensured that LPS+IL-10 does boost Arginase-2 synergistically in human macrophages (**Fig. 1e-h and S1d, line 77-81**).*

*Furthermore, we have included new data using the LPS-induced lethality model in WT vs Arg2 deficient mice (**Fig. 4m-n, and S4j**) where we now have a more valid disease context in relation to Arg2. Bacterial LPS has been extensively used in models studying inflammation as it mimics many inflammatory effects of cytokines, such as TNF, IL-1 β or IL-6. Consequently, the LPS model has been reported to be most suitable when being interested in the impact of new therapies for acute inflammation and for understanding the systemic inflammatory response (<https://doi.org/10.1186/s12929-017-0370-8>). We hope for the purposes of this study that this model is sufficient to validate the importance of Arg2 in a disease setting.*

Some results are a replication of Ip et al. paper, which takes away the novelty factor.

The reviewer makes a very valid point that the novelty factor is lost somewhat with the replication of the IL-10^{-/-} OxPhos experiment (Fig 3 in our original manuscript) by Ip et al., in our hands. We have now moved this to the supplemental section (Fig. S3a-b).

Metabolomics and ETC results in Figure 4 are not interpreted correctly and the presented data do not support the conclusions due to lacking experimental conditions. Specifically: 1. Exogenous addition of IL-10 to LPS-stimulated macrophages M(LPS+IL10): LPS will cause an increase in endogenous IL-10 and the further addition of IL-10 seems artificial, without mimicking any *in vivo* state of activation of macrophages. I understand that the usage of IL10^{-/-} mice controls for the endogenous IL-10, and the authors have a previous paper on miR-155/IL-10 pathway. However, the specific conditions that trigger Arg2 in BMDMs (LPS +IL-10) are context and (possibly) species-specific. The manuscript will benefit from acknowledging and discussing this throughout. This is important as most of the significant results on the mitochondrial effect of Arg2 (e.g. mitochondrial fusion, membrane potential) are obtained using this specific condition.

We thank the reviewer for highlighting a very relevant point. However, we are confident that the LPS+IL-10 stimulation that enhances Arg2 expression and subsequently modulates mitochondrial bioenergetics is a real effect.

Firstly, we have amended the introduction to clarify that LPS stimulation results in autocrine IL-10 production (lines 49-50), which then regulates excess inflammatory effects initiated by LPS. Given our time point of stimulation is 24 hours, it can be difficult to observe IL-10 specific effects in WT macrophages at that late time point. Hence, the use of exogenous IL-10 enhances the effects already initiated by autocrine IL-10. As many of our key experiments that showed Arg-2 relevant effects are IL-10 dependent, we weren't able to observe differences between WT and Arg2^{-/-} in LPS-only treated cells e.g. succinate and fumarate assays (Fig. 4h-i), the SDH-activity spectrophotometric (Fig.4g) and MTT assays (Fig. 4f), the IL-1b ELISA (Fig.4l).

Secondly, as the reviewer noted themselves, we have shown using IL-10 deficient BMDM (Fig. 1j) and the use of STAT3 inhibitor, the dependency of IL-10 on Arg2 modulation (Fig S1e and S1g).

Furthermore, the use of human macrophages, as suggested by the reviewer, has added to our studies and confirmed that this effect is also observed in human cells (Fig. 1e-h and S1d, line 77-81).

Finally, we would like to point out that our stimulation of LPS+IL-10 was based on the findings of many other well-known papers in the IL-10 field, such as data from Ip et al., (2017) and Murray et al.,(2002) and those listed below:

References:

Ip, W. K. E., Hoshi, N., Shouval, D. S., Snapper, S. & Medzhitov, R. (2017). Anti-inflammatory effect of IL-10 mediated by metabolic reprogramming of macrophages. *Science* **356**, 513-519, doi:10.1126/science.aal3535

Conaway, E.A, De Oliveira, D.C, McInnis, C.M, Snapper, S.B and Horwitz, B. (2017). Inhibition of Inflammatory Gene Transcription by IL-10 Is Associated with Rapid Suppression of Lipopolysaccharide-Induced Enhancer Activation. *Journal of Immunology*. **198** (7), 2906-2915.

Murray, P.J. (2005). The primary mechanism of the IL-10-regulated antiinflammatory response is to selectively inhibit transcription. *PNAS*. **102** (24), 8686-8691.

Lang, R., Patel, D., Morris, J. J., Rutschman, R. L. & Murray, P. J. (2002). Shaping gene expression in activated and resting primary macrophages by IL-10. *Journal of Immunology*. **169**, 2253-2263, doi:10.4049/jimmunol.169.5.2253

Inoue, G. (2000). Effect of interleukin-10 (IL-10) on experimental LPS-induced acute lung injury. *Journal of Infection and Chemotherapy*. **6** (1), 51-60.

2. Were the Arg1+/- mice viable and showed decreased Arg1 and urea in M(LPS+IL-10)?

Unfortunately, we were unaware of the viability of Arg1+/- mice and did not have access or time to include these mice in experiments for the reviewed manuscript. However, it is really good to know they are viable and will work to include them in our future studies.

Moreover, we believe that siRNA and overexpression studies show a clear difference between Arg1 and Arg2 isoforms (Fig.3g-h and S4i). We have included extra data (Arg1 overexpression and CII OCR activity: Fig.4e) to really highlight that the Arg2 isoform is most important for mitochondrial effects.

3. Figure 2b convincingly shows the mitochondrial localization of Arg2 in BMDMs stimulated with LPS+IL-10. Is the same true in human macrophages?

To date no one has shown mitochondrial localization or upregulation of Arg2 in human macrophages in response to LPS+IL-10. We performed additional experiments and demonstrate the significant upregulation of Arg2 in LPS+IL-10 stimulated THP-1 cells (Fig.1 g-h). The same upregulation was not observed for Arg1 in human macrophages. (Fig.1 h and S1d). Unfortunately, due to covid19 restrictions, we were unable to process enough cells for mitochondrial fractionation of human macrophages in time.

4. What is the effect of mir-155 on the mitochondrial effects of Arg2?

We can confirm that miR-155 deficiency enhances the expression of Arg2 as shown by the western in Fig1d and at the mitochondria as shown here:

We are currently working on another manuscript of Arg2 regulation via miR155, where we show that boosting Arg2 by inhibition of its miR-155 binding site boosted oxidative respiration, and downregulated proinflammatory cytokine production. We aim to publish those findings in the near future.

5. Figure 3a-b is a replication of previously published results by Ip et al. (reference 18). The siRNA experiment was done in Raw cells and should be replicated in BMDMs for accurate comparison with the Arg2 KO mice BMDMs.

This comment regarding the replication of data by Ip et al., is in agreement with the point made by Reviewer 2. In agreement, we have now moved Figures 3a-b from the original manuscript to supplemental and this data now makes up Fig. S3a-b in the revised manuscript.

Extensive efforts were made to conduct siRNA experiments in primary BMDMs. This was included but not limited to employing: NEON electroporation; Dharmacon SMART RNA pool using Lipofectamine 3000; Accell siRNA (Dharmacon). Moreover, these BMDM were no longer viable after transfection to reliably perform Seahorse Flux Analyzer experiments

In agreement with the reviewer for a closer comparison to Arg2^{-/-} BMDM we performed siRNA experiments in immortalised BMDM (iBMDM). These experiments are represented now in Fig. 3a-b, S3d and S4i and confirms our original findings in Raw cells.

6. Figure 4a: If fumarate and malate levels are the same between LPS and LPS+IL-10 conditions, then it would not be correct to claim that LPS+IL-10 is causing a specific metabolic shift. Statistics should be done between LPS and LPS+10 and in case it is not significant, the results should be re-considered.

Work from Baseler et al., demonstrated that the LPS-induced accumulation in succinate in wild-type BMDM was abolished in IL-10^{-/-} BMDM, yet the reductions of alpha-ketoglutarate, indicative of the LPS-induced "TCA break", remained completely intact. Through that result, they suggested that "autocrine IL-10 may act to relieve the TCA cycle "break" by restraining succinate levels", and increasing downstream metabolites such as fumarate and malate. These observations were validated in our GC/MS data where we saw that addition of exogenous IL-10 (i.e. LPS+IL-10) resulted in a further increase in these metabolites compared to LPS alone (Fig S4a). Of course, it was difficult to tease apart the exact contribution of IL-10 in

a WT BMDM (due to effects of autocrine IL-10 being produced in LPS-only treated macrophages), hence the significance between LPS and LPS+IL-10 could not be seen. We circumvented this issue then by comparing significance between unstimulated samples vs LPS, and unstimulated samples vs LPS+IL-10. LPS+IL-10 samples gave us significant increases in fumarate and malate, but not succinate. LPS-alone samples didn't give us that significance.

Interestingly as well, when we looked at cellular succinate and fumarate levels, we could clearly see that in a WT BMDM, succinate was accumulated with LPS but reduced with LPS+IL-10, whereas fumarate was decreased with LPS, and significantly increased with LPS+IL-10, an effect we showed to be dependent on Arg2 (**Fig. 4h-i**). This led us to surmise that IL-10 is working to downregulate excess (inflammatory) succinate levels by boosting fumarate levels through Arg2. We currently perceive that this is at least partly due to increased SDH activity mediated via Arg2. Future work will need to validate these findings by looking at other non-mitochondrial sources of succinate/fumarate in WT and Arg2 deficient BMDM using tracing experiments.

Reference:

Baseler, W. A. *et al.* Autocrine IL-10 functions as a rheostat for M1 macrophage glycolytic commitment by tuning nitric oxide production. *Redox biology* **10**, 12-23, doi:10.1016/j.redox.2016.09.005 (2016)

7. Figure 4b: The LPS condition should be added – the comparison between LPS+IL-10 and CTRL is only partially informative. Same with Figure 4e.

We have now added LPS-only conditions for these figures represented in Fig 4a,c.

8. Figure 4g. The over-expression of Arg1 should also be investigated in terms of CII activity.

We thank the reviewer for this suggestion and the overexpression of Arg1 has been included in additional CII activity experiments. Overexpression of Arg1 resulted in no significant change in CII activity compared to empty vector (EV) controls (Fig. 4e)

9. The in vivo data (Figure 4k-m) is premature and show only mRNA levels of cytokines. Figure 4k shows IL-1beta mRNA levels (right panel) whereas Figure 4l also shows mRNA levels of IL-1beta but this time significant. This part of the paper is significantly weak and needs to be reconsidered in view of an appropriate in vivo model such as colitis model in Il10^{-/-} mice.

We acknowledge that our previous in vivo data in IL-10 deficient mice didn't provide novel insights other than how the expression of Arg2 varied in IL-10 deficient mice compared to wild-types. Whilst we completely agree that the colitis model would be very interesting to pursue, we felt it would warrant a much more in depth study surrounding the role of Arg2 in that particular setting.

Hence, in order to strengthen this aspect of the manuscript, we used the LPS-induced model of inflammation in Arg2 deficient mice. We continued the use of the LPS in vivo model in this study as it captures important inflammatory and signalling responses in many autoimmune and inflammatory diseases. These new experiments with Arg2 deficient mice have added novelty and disease relevance as

we showed significantly increased HIF-1 α and IL-1 β levels in the spleen and peritoneal lavage, respectively (Fig. 4m-n). What was particularly interesting was that this model showed us that despite increased IL-10 levels in the peritoneal lavage in the Arg2 knock-outs, IL-1 β levels were significantly higher in the knock-outs compared to wild-type mice. This mirrored our ex vivo and in vitro experiments that also showed enhanced IL-1 β in Arg2 deficient macrophages despite presence of IL-10 (Fig. 4I and S4i).

REVIEWERS' COMMENTS

Reviewer #1 (Remarks to the Author):

In my opinion, the authors have addressed the major (and many minor) concerns raised

Reviewer #2 (Remarks to the Author):

The authors have addressed several of my concerns and have improved the data presentation and discussion. Importantly, they provide new data from three assays to argue that Arg2 regulates SDH activity. The results substantiate their complex II-dependent respiration experiments and demonstrate that Arg2 positively regulates SDH activity. Unfortunately, however, the mechanism how Arg2 promotes SDH activity remains largely unclear and therefore the importance of the localization of Arg2 to mitochondria unexplained.

I agree with the authors that they now have more evidence that Arg2, and not Arg1, regulate mitochondrial function. However, it is a pity that the delta-MTS Arg2 experiment could not be performed. Therefore, concerns about the localization of Arg2 remain and the interpretation of Arg2 mitochondrial localisation remains unconvincing. The authors performed a new mitochondrial import experiment (Fig. 2a) and found that mitochondrial import of Arg2 to be independent of membrane potential, "suggesting that Arg2 is localized in the inner mitochondrial membrane". It is surprising that the import of an MTS-containing protein is independent of membrane potential and I would encourage the authors to avoid suggesting the localization of Arg2 to be in the inner membrane if it does not contain a transmembrane domain. In Fig. 2a, please clarify which samples were treated with proteinase K. As presented, the data do not convincingly show any Arg2 import into mitochondria. Please also include Mw labels in Fig 2a.

I still don't agree that Fig 2b shows Arg2 "is upregulated at the mitochondria in response to LPS+IL-10 compared to LPS or IL-10 alone" without quantification. I think the effect is more striking in the IL10-/- fractions included in the rebuttal. Does this not show more convincingly that Arg2 is localized to the mitochondrial fraction in response to LPS and IL-10 treatment? I would encourage the authors to fit this entire figure in, especially since the import experiments don't address Arg2 import upon LPS+IL10 treatment. Importantly, the purpose of running mitochondrial and cytosolic fractions on the same gel is to gain an accurate impression of relative protein abundance in both compartments by ensuring that both fractions are blotted, probed and exposed in the same manner.

Reviewer #3 (Remarks to the Author):

In their revised manuscript, the authors added new data that address all the initial concerns – especially the human relevance of the findings and the lack of in vivo model. The authors should be congratulated for these exciting findings and their solid rebuttal.

Jacques Behmoaras (Imperial College London)

REVIEWERS' COMMENTS

We wish to thank the reviewers again for their helpful input on our revised manuscript. Please find specific comments to any concerns remaining below:

Reviewer #1 (Remarks to the Author):

In my opinion, the authors have addressed the major (and many minor) concerns raised.

We are delighted that we were able to address both the major and minor concerns raised.

Reviewer #2 (Remarks to the Author):

The authors have addressed several of my concerns and have improved the data presentation and discussion. Importantly, they provide new data from three assays to argue that Arg2 regulates SDH activity. The results substantiate their complex II-dependent respiration experiments and demonstrate that Arg2 positively regulates SDH activity. Unfortunately, however, the mechanism how Arg2 promotes SDH activity remains largely unclear and therefore the importance of the localization of Arg2 to mitochondria unexplained. I agree with the authors that they now have more evidence that Arg2, and not Arg1, regulate mitochondrial function. However, it is a pity that the delta-MTS Arg2 experiment could not be performed. Therefore, concerns about the localization of Arg2 remain and the interpretation of Arg2 mitochondrial localisation remains unconvincing.

We are pleased that the reviewer found our additional SDH-based assays satisfactory. We acknowledge that more work needs to be undertaken in order to fully appreciate the complexities of Arg2-SDH interactions. Taking the reviewer's comments on board, we have now included a limitations section at the end of our discussion. Future work aims to delineate the role of Arg2 specifically at the mitochondria by performing in-depth metabolomics studies in Arg2 deficient mice.

The authors performed a new mitochondrial import experiment (Fig. 2a) and found that mitochondrial import of Arg2 to be independent of membrane potential, "suggesting that Arg2 is localized in the inner mitochondrial membrane". It is surprising that the import of an MTS-containing protein is independent of membrane potential and I would encourage the authors to avoid suggesting the localization of Arg2 to be in the inner membrane if it does not contain a transmembrane domain. In Fig. 2a, please clarify which samples were treated with proteinase K. As presented, the data do not convincingly show any Arg2 import into mitochondria. Please also include Mw labels in Fig 2a.

Indeed, it is peculiar that an MTS protein is imported irrespective of membrane potential. However, we used a high dose of FCCP (25 μ M), and Arg2 was still present which leads us to merely speculate that it may be present at the IMM. However, to satisfy the reviewer's comments we have amended the text to downplay the claim about Arg2's sub-localization in the mitochondria, indicating that future studies are required to delineate its exact localization.

In addition, we have now clarified within the revised manuscript that none of the samples were treated with Proteinase K in the time-course experiment. MW labels have also been included in both the main manuscript Figure and in the Source Data File for Figure 2a.

I still don't agree that Fig 2b shows Arg2 "is upregulated at the mitochondria in response to LPS+IL-10 compared to LPS or IL-10 alone" without quantification. I think the effect is more striking in the IL10-/- fractions included in the rebuttal. Does this not show more convincingly that Arg2 is localized to the mitochondrial fraction in response to LPS and IL-10 treatment? I would encourage the authors to

fit this entire figure in, especially since the import experiments don't address Arg2 import upon LPS+IL10 treatment. Importantly, the purpose of running mitochondrial and cytosolic fractions on the same gel is to gain an accurate impression of relative protein abundance in both compartments by ensuring that both fractions are blotted, probed and exposed in the same manner.

In agreement with the reviewer about the LPS+IL-10 effect, we have now swapped to the IL-10^{-/-} fractions in the main manuscript (fig 2b) but the full fractionation blots will also be available in supplementary information source_data_file.

Reviewer #3 (Remarks to the Author):

In their revised manuscript, the authors added new data that address all the initial concerns – especially the human relevance of the findings and the lack of in vivo model. The authors should be congratulated for these exciting findings and their solid rebuttal.

Jacques Behmoaras (Imperial College London)

We are grateful to the reviewer for their compliments on our revised script.